# THE NOISE GEOMETRY OF STOCHASTIC GRADIENT DESCENT: A THEORETICAL AND QUANTITATIVE CHARACTERIZATION

## ABSTRACT

Empirical studies have demonstrated that the noise in stochastic gradient descent (SGD) aligns favorably with the local geometry of loss landscape. However, theoretical and quantitative explanations for this phenomenon remain sparse. In this paper, we offer a comprehensive theoretical investigation into the aforementioned *noise geometry* for over-parameterized linear (OLMs) models and two-layer neural networks. We scrutinize both average and directional alignments, paying special attention to how factors like sample size and input data degeneracy affect the alignment strength. As a specific application, we leverage our noise geometry characterizations to study how SGD escapes from sharp minima, revealing that the escape direction has significant components along flat directions. This is in stark contrast to GD, which escapes only along the sharpest directions. To substantiate our theoretical findings, both synthetic and real-world experiments are provided.

## 1 INTRODUCTION

Stochastic gradient descent (SGD) and its variants have become the de facto optimizers for training machine learning models (Bottou, 1991). Unlike full-batch gradient descent (GD), SGD uses only mini-batches of data in each iteration, which injects noise into the optimization process. This noise can have a pronounced impact on both the convergence behavior (Thomas et al., 2020; Wojtowytsch, 2023; Feng and Tu, 2021; Simsekli et al., 2019) and the generalization capabilities (Zhang et al., 2017; Keskar et al., 2017; Wu et al., 2017; Zhu et al., 2019; Smith et al., 2020) of the algorithm.

Zhu et al. (2019); Wu et al. (2020); Xie et al. (2020) showed that SGD noise is highly anisotropic and in particular, the noise covariance matrix aligns well with the Hessian matrix. As such, they propose a Hessian-based approximation of the noise covariance: $\Sigma(\boldsymbol{\theta}) \approx \sigma^2 H(\boldsymbol{\theta})$, where $\Sigma(\boldsymbol{\theta})$ and $H(\boldsymbol{\theta})$ denote the noise covariance and Hessian matrices at $\boldsymbol{\theta}$, respectively and $\sigma$ serves as a small constant denoting the noise magnitude. Subsequent works (Feng and Tu, 2021; Mori et al., 2022; Wojtowytsch, 2021; Liu et al., 2021) presented an improved Hessian-based approximation: $\Sigma(\boldsymbol{\theta}) \approx 2L(\boldsymbol{\theta})H(\boldsymbol{\theta})$ for regression problems with square loss, where $L(\boldsymbol{\theta})$ denotes the loss value. This refined approximation acknowledges the fact that the noise magnitude is proportional to the loss value.

However, the alignment between SGD noise and local landscape geometry remains empirical observations, lacking quantitative characterization and theoretical grounding. Hessian-based approximations are not accurate, as underscored by Thomas et al. (2020). A recent effort by Wu et al. (2022) employed a normalized cosine similarity between $\Sigma(\boldsymbol{\theta})$–which is close to the Hessian matrix in low loss regions–and the empirical Fisher matrix $G(\boldsymbol{\theta})$ as a metric to quantify the alignment. This metric is inspired by analyzing the dynamical stability of SGD (Wu et al., 2018) and can be interpreted as certain type of average alignment. Nevertheless, the analysis in Wu et al. (2022) is restricted to over-parameterized linear models (OLMs) and operates under the assumption of infinite data, leaving open questions about the generalizability of such alignment in more practically relevant settings.

**Our contribution.** Let $n, d$ denote the sample size, input dimension, respectively. Then, our contributions can be summarized as follows.

- We first extend the average alignment analysis (Wu et al., 2022) to finite sample scenarios, offering a comprehensive investigation of how factors like sample size and input data degeneracy impact the alignment strength. We establish that, as long as $d_{\text{eff}} \gtrsim \log n$, the alignment strength

is lower-bounded for both OLMs and two-layer neural networks–models not considered in Wu et al. (2022). Here, $d_{\text{eff}}$ represents an effective input dimension, and this condition accommodates the important regimes like $n \sim \log(d_{\text{eff}})$ (for sparse recovery) and $n \sim d_{\text{eff}}$ (the proportional scaling).

- We then delve into a directional alignment analysis, probing whether the component of noise energy along a specific direction is proportional to the curvature in that direction. Our results show that for OLMs, as long as $n \gtrsim d$, the strength of directional alignment is lower-bounded acorss all directions and the entire parameter space.

- Lastly, we provide a detailed analysis of the mechanisms by which SGD escapes from sharp minima by leveraging our noise geometry results. We show that *the escape direction of SGD exhibits significant components along flat directions of the local landscape*. This stands in stark contrast to GD, which escapes from minima only along the sharpest direction. We also discuss the implications of this unique escape behavior, providing a preliminary explaination of how cyclical learning rate (Smith, 2017; Loshchilov and Hutter, 2017) can help find flatter minima.

It is worh noting that our theoretical guarantees apply effectively to both isotropic and anisotropic inputs, and *the guaranteed alignment strength is independent of the degree of overparameterization*. In addition, all theoretical findings are supported by numerical experiments conducted on both small-scale and larger-scale models. To justify the practical relevance, experiments of classifying CIFAR-10 dataset using VGG nets and ResNets are also provided in Section 6. Overall, our work advances the theoretical understanding of the geometry of SGD noise and provides insights into how SGD navigates the loss landscape.

## 1.1 OTHER RELATED WORK

**Noise geometry.** Ziyin et al. (2022) provides a detailed analysis of the noise structure of online SGD for linear regression. We instead consider nonlinear models and finite-sample regimes. We also acknowledge the existence of works such as Simsekli et al. (2019); Zhou et al. (2020), which argue that the magnitude of SGD noise is heavy-tailed. However, our particular focus is on the noise shape and the observation that the noise magnitude is directly proportional to the loss value.

**Escape from minima and saddle points** The phenomenon of SGD escaping from sharp minima exponentially fast was initially studied in Zhu et al. (2019) as an indicator of how much SGD dislikes sharp minima. This provides an explanation of the famous "flat minima hypothesis" (Hochreiter and Schmidhuber, 1997; Keskar et al., 2017; Wu and Su, 2023)—one of the most important observations in explaining the implicit regularization of SGD. However, existing analyses of the escape phenomenon have primarily focused on the escape rate (Wu et al., 2018; Zhu et al., 2019; Xie et al., 2020; Mori et al., 2022; Ziyin et al., 2022). In contrast, we extends this focus by providing analysis of escape direction, which is enabled by our characterizations of the noise geometry. Kleinberg et al. (2018) introduced an alternative perspective, positing that SGD circumvents local minima by navigating an effective loss landscape that results from the convolution of the original landscape with SGD noise. In this context, our noise geometry characterizations can be beneficial in understanding the effective loss landscape. In addition, prior works like (Daneshmand et al., 2018; Xie et al., 2022) has illustrated that the alignment of noise with local geometry facilitates the rapid saddle-point escape of SGD. Our work offers theoretical substantiation for the alignment assumptions in these studies.

## 2 PRELIMINARIES

**Notation.** We use bold letters for vectors and lowercase letters for scalars, e.g. $\boldsymbol{x} = (x_1, \cdots, x_d)^\top$. We use $\langle \cdot, \cdot \rangle$ for the Euclidean inner product and $\|\cdot\|_p$ for the $l_p$ norm of a vector or the spectral norm of a matrix. Denote by $\mathcal{N}(\boldsymbol{\mu}, S)$ the Gaussian distribution with mean $\boldsymbol{\mu}$ and covariance matrix $S$, while we define $\mathbb{U}(\Omega)$ as the uniform distribution on a set $\Omega$. For a matrix $A$, we refer to its eigenvalues in a decreasing order as $\{\lambda_j(A)\}_j$. For a positive definitive matrix $A$, we use $\text{cond}(A) := \lambda_{\max}(A)/\lambda_{\min}(A)$ and $\text{srk}(A) := \text{Tr}(A)/\|A\|_2$ to denote the condition number and the stable rank of $A$, respectively. We use $a \lesssim b$ to mean there exist an an absolute constant $C > 0$ such that $a \leq Cb$ and $a \gtrsim b$ is defined analogously. We write $a \sim b$ if there exist absolute constants $C_1, C_2 > 0$ such that $C_1 b \leq a \leq C_2 b$.

Let $\{(\boldsymbol{x}_i, y_i)\}_{i=1}^n \subset \mathbb{R}^d \times \mathbb{R}$ be the training set and $f(\cdot; \boldsymbol{\theta}) : \mathbb{R}^d \to \mathbb{R}$ be the model parameterized by $\boldsymbol{\theta} \in \mathbb{R}^p$. Let $\ell_i(\boldsymbol{\theta}) = \frac{1}{2}(f(\boldsymbol{x}_i; \boldsymbol{\theta}) - y_i)^2$ be the square loss at the $i$-th sample and $\mathcal{L}(\boldsymbol{\theta}) =$

$\frac{1}{n}\sum_{i=1}^{n}\mathcal{L}_i(\boldsymbol{\theta})$ be the empirical risk. To minimize $\mathcal{L}(\cdot)$, the mini-batch SGD updates as follows

$$\boldsymbol{\theta}(t+1) = \boldsymbol{\theta}(t) - \frac{\eta}{B}\sum_{i\in\mathcal{B}_t}\nabla\ell_i(\boldsymbol{\theta}(t)), \tag{1}$$

where $\mathcal{B}_t = \{\gamma_{t,1},\cdots,\gamma_{t,B}\}$ is a batch with size $|\mathcal{B}_t| = B$, and $\gamma_{t,1},\cdots,\gamma_{t,B} \stackrel{\text{i.i.d.}}{\sim} \mathbb{U}([n])$.

To isolate the impact of noise, the SGD update (1) is often reformulated as follows

$$\boldsymbol{\theta}(t+1) = \boldsymbol{\theta}(t) - \eta\left(\nabla\mathcal{L}(\boldsymbol{\theta}(t)) + \boldsymbol{\xi}(t)\right), \tag{2}$$

where $\nabla\mathcal{L}(\boldsymbol{\theta}(t))$ is the full-batch gradient and $\boldsymbol{\xi}(t)$ represents the mini-batch noise satisfying $\mathbb{E}[\boldsymbol{\xi}(t)] = 0, \mathbb{E}[\boldsymbol{\xi}(t)\boldsymbol{\xi}(t)^\top] = \Sigma(\boldsymbol{\theta}(t))/B$ with the noise covariance given by

$$\Sigma(\boldsymbol{\theta}) = \frac{1}{n}\sum_{i=1}^{n}\nabla\ell_i(\boldsymbol{\theta})\nabla\ell_i(\boldsymbol{\theta})^\top - \nabla\mathcal{L}(\boldsymbol{\theta})\nabla\mathcal{L}(\boldsymbol{\theta})^\top. \tag{3}$$

In the above setup, the Hessian matrix of the empirical risk is given by

$$H(\boldsymbol{\theta}) = G(\boldsymbol{\theta}) + \frac{1}{n}\sum_{i=1}^{n}\left(f(\boldsymbol{x}_i;\boldsymbol{\theta}) - y_i\right)\nabla^2 f(\boldsymbol{x}_i;\boldsymbol{\theta}), \tag{4}$$

where $G(\boldsymbol{\theta}) = \frac{1}{n}\sum_{i=1}^{n}\nabla f(\boldsymbol{x}_i;\boldsymbol{\theta})\nabla f(\boldsymbol{x}_i;\boldsymbol{\theta})^\top$ is the empirical Fisher matrix. Eqn. (4) implies that when the fit errors are small, we have $G(\boldsymbol{\theta}) \approx H(\boldsymbol{\theta})$ and in particular, for global minima $\boldsymbol{\theta}^*$, $H(\boldsymbol{\theta}^*) = G(\boldsymbol{\theta}^*)$. Additionally, for linear regression $f(\boldsymbol{x};\boldsymbol{\theta}) = \boldsymbol{\theta}^\top\boldsymbol{x}$, $H(\boldsymbol{\theta}) = G(\boldsymbol{\theta}) \equiv \frac{1}{n}\sum_{i=1}^{n}\boldsymbol{x}_i\boldsymbol{x}_i^\top$.

**Over-parameterized linear models (OLMs).** An OLM is defined as $f(\boldsymbol{x};\boldsymbol{\theta}) = F(\boldsymbol{\theta})^\top\boldsymbol{x}$, where $F : \mathbb{R}^p \to \mathbb{R}^d$ denotes a general re-parameterization function. Although $f(\cdot;\boldsymbol{\theta})$ only represents linear functions, the corresponding loss landscape can be highly non-convex. Some typical examples include (i) the linear model $F(\boldsymbol{w}) = \boldsymbol{w}$; (ii) the diagonal linear network: $F(\boldsymbol{\theta}) = (\alpha_1^2 - \beta_1^2, \ldots, \alpha_d^2 - \beta_d^2)^\top$; and (iii) the linear network: $F(\boldsymbol{\theta}) = W_1 W_2 \cdots W_L$. Notably, OLMs have been widely used to analyze the optimization and implicit bias of SGD (Arora et al., 2019; Woodworth et al., 2020; Pesme et al., 2021; HaoChen et al., 2021; Azulay et al., 2021).

**Noise Geometry.** Before proceeding to our refined characterization of the noise geometry, we first recall two existing results on quantifying the geometry of SGD noise.

- Mori et al. (2022) proposed the following Hessian-based approximation:

$$\Sigma(\boldsymbol{\theta}) \approx 2\mathcal{L}(\boldsymbol{\theta})G(\boldsymbol{\theta}). \tag{5}$$

  It reveals 1) the noise magnitude is proportional to the loss value; 2) the noise covariance aligns with the Fisher matrix. This approximation is intuitive and helpful for understanding, but it cannot be accurate in general.

- *Online SGD for OLMs with Gaussian inputs.* Suppose $\boldsymbol{x} \sim \mathcal{N}(\boldsymbol{0}, S)$ and $n = \infty$ (i.e., online SGD). For OLMs, Wu et al. (2022) derived the following analytical expression

$$\Sigma(\boldsymbol{\theta}) = 2\mathcal{L}(\boldsymbol{\theta})G(\boldsymbol{\theta}) + \nabla\mathcal{L}(\boldsymbol{\theta})\nabla\mathcal{L}(\boldsymbol{\theta})^\top. \tag{6}$$

  In this case, the approximation (5) fails to capture the extra rank-1 term.

## 3 AVERAGE ALIGNMENT

Let $\Sigma_1(\boldsymbol{\theta}) = \frac{1}{n}\sum_{i=1}^{n}\nabla\ell_i(\boldsymbol{\theta})\nabla\ell_i(\boldsymbol{\theta})^\top, \Sigma_2(\boldsymbol{\theta}) = \nabla\mathcal{L}(\boldsymbol{\theta})\nabla\mathcal{L}(\boldsymbol{\theta})^\top$. Then $\Sigma(\boldsymbol{\theta}) = \Sigma_1(\boldsymbol{\theta}) - \Sigma_2(\boldsymbol{\theta})$. Following Wu et al. (2022), we consider the following metrics of quantifying average alignment:

$$\tilde{\mu}(\boldsymbol{\theta}) = \frac{\text{Tr}\left(\Sigma(\boldsymbol{\theta})G(\boldsymbol{\theta})\right)}{2\mathcal{L}(\boldsymbol{\theta})\|G(\boldsymbol{\theta})\|_{\text{F}}^2}, \quad \mu(\boldsymbol{\theta}) := \frac{\text{Tr}(\Sigma_1(\boldsymbol{\theta})G(\boldsymbol{\theta}))}{2\mathcal{L}(\boldsymbol{\theta})\|G(\boldsymbol{\theta})\|_{\text{F}}^2}. \tag{7}$$

It is commonly believed that the magnitude of the full-batch gradient $\nabla\mathcal{L}$ is relatively small compared to the sample gradients $\{\nabla\ell_i\}_i$. Consequently, the influence of $\Sigma_2(\boldsymbol{\theta})$ would be negligible compared to $\Sigma_1(\boldsymbol{\theta})$ and thus, $\tilde{\mu}(\boldsymbol{\theta})$ and $\mu(\boldsymbol{\theta})$ often behave similarly. Specifically, Wu et al. (2022) has provably demonstrated that the difference between $\tilde{\mu}(\boldsymbol{\theta})$ and $\mu(\boldsymbol{\theta})$ is neglibile in terms of controling the dynamical stability of SGD. We refer to Wu et al. (2022) for more details about the difference. Thus, we only focus on studying $\mu(\cdot)$ in this section.

## 3.1 OVER-PARAMETERIZED LINEAR MODELS

The analytical expression (6) guarantees $\mu(\boldsymbol{\theta}) \geq 1$ in an infinite data scenario. The following theorem extends it to finite-sample cases and the proof can be found in Appendix B. To simplify the statement, we define the effective dimension of inputs as follows

$$d_{\text{eff}} := \min\{\text{srk}(S), \text{srk}(S^2)\},$$

where $S$ represents the input covariance matrix and $\text{srk}(S) = \text{tr}(S)/\|S\|_2$ is the stable rank of $S$. In particular, when $S$ is isotropic, we have $d_{\text{eff}} = d$.

**Theorem 3.1.** *Consider OLMs and assume $\boldsymbol{x}_1, \boldsymbol{x}_2, \ldots, \boldsymbol{x}_n \overset{\text{i.i.d.}}{\sim} \mathcal{N}(\mathbf{0}, S)$. For any $\epsilon, \delta \in (0, 1)$,*

(a) *if $n/\log(n/\delta) \gtrsim 1/\epsilon^2$ and $d_{\text{eff}} \gtrsim \log(n/\delta)/\epsilon^2$, then w.p. at least $1 - \delta$, it holds that $\inf_{\boldsymbol{\theta} \in \mathbb{R}^p} \mu(\boldsymbol{\theta}) \geq \frac{(1-\epsilon)^2}{(1+\epsilon)^2 \text{cond}^2(\nabla F(\boldsymbol{\theta}) \nabla F(\boldsymbol{\theta})^\top)}$;*

(b) *if $n \gtrsim d + \log(1/\delta)$, then w.p. at least $1 - \delta$, it holds that $\inf_{\boldsymbol{\theta} \in \mathbb{R}^p} \mu(\boldsymbol{\theta}) \gtrsim 1$.*

Result (a) is established by leveraging the high dimensionality of inputs, as stated by the condition $d_{\text{eff}} \gtrsim \log n$, which is particularly relevant for low-sample regimes. Notably, this includes the important regimes like $n \sim \log(d_{\text{eff}})$ (for sparse recovery) and $n \sim d_{\text{eff}}$ (the proportional scaling). In contrast, result (b) is pertinent to the enough-data regime where $n \gtrsim d$. Notably, the alignment holds no matter how degenerate the covariance matrix is. This is obtained by scrutinizing the concentration around the population alignment as characterized in equation (6). In a summary, these two results are complementary and collectively span all the regimes of interest.

**Example.** Consider the isotropic case where $S = I_d$ and linear regression $F(\boldsymbol{w}) = \boldsymbol{w}$. In this case, $\nabla F(\boldsymbol{w}) \equiv I_d$ and thus, Theorem 3.1 implies that it holds that $\inf_{\boldsymbol{\theta} \in \mathbb{R}^p} \mu(\boldsymbol{\theta}) \gtrsim 1$ as long as $n \gtrsim 1$.

*Remark* 3.2. We would like to emphasize that the conditions presented in Theorem 3.1 are independent of the model size $p$. Consequently, these alignment results can be effectively applied to linear networks regardless of their width and depth.

## 3.2 TWO-LAYER NEURAL NETWORKS

Consider two-layer neural networks given by $f(\boldsymbol{x}; \boldsymbol{\theta}) = \sum_{k=1}^m a_k \phi(\boldsymbol{b}_k^\top \boldsymbol{x})$ with $a_k \in \{\pm 1\}$ to be fixed. We use $\boldsymbol{\theta} = (\boldsymbol{b}_1^\top, \cdots, \boldsymbol{b}_m^\top)^\top \in \mathbb{R}^{md}$ to denote the concatenation of all trainable parameters. Here, $\phi : \mathbb{R} \mapsto \mathbb{R}$ is an activation function with a nondegenerate derivative as defined below.

**Assumption 3.3.** There exist constants $\beta > \alpha > 0$ such that $\alpha \leq \phi'(z) \leq \beta$ holds for any $z \in \mathbb{R}$.

**Example 3.4.** *(i) A typical activation function that satisfies Assumption 3.3 is $\alpha$-Leaky ReLU: $\phi(z) = \max\{\alpha z, z\}$, where $\alpha \in (0, 1)$. (ii) Moreover, the assumption also holds for Sigmoid with the truncation trick (to prevent gradient vanishing of Sigmoid): $\phi(z) = 1/(1 + \exp(-\text{sgn}(z) \min\{|z|, M\}))$, where $M > 0$ is the truncation constant.*

**Theorem 3.5.** *Consider the two-layer network $f(\cdot; \boldsymbol{\theta})$ with the activation function satisfying Assumption 3.3 and assume $\boldsymbol{x}_1, \cdots, \boldsymbol{x}_n \overset{\text{i.i.d.}}{\sim} \mathcal{N}(\mathbf{0}, S)$. For any $\epsilon, \delta \in (0, 1)$, if $n/\log(n/\delta) \gtrsim 1/\epsilon^2$ and $d_{\text{eff}} \gtrsim \log(n/\delta)/\epsilon^2$, then w.p. at least $1 - \delta$, it holds that $\inf_{\boldsymbol{\theta} \in \mathbb{R}^{md}} \mu(\boldsymbol{\theta}) \geq \frac{\alpha^2(1-\epsilon)^2}{\beta^2(1+\epsilon)^2}$.*

This theorem establishes a uniform lower bound for the alignment strength, quantified by $\mu(\boldsymbol{\theta})$. Importantly, the number of samples required remains independent of the network width $m$. The proof follows a similar approach to that of Theorem 3.1 and can be found in Appendix B. Note that we impose two specific conditions: the activation gradient must be non-degenerate and the output-layer coefficients are non-trainable. We stress that these conditions are obligatory soly for establishing alignment across the *entire loss landscape*. In practice, such stringent conditions may not be necessary, as the focus is on regions navigated by SGD. Figure 1b corroborates that alignment is indeed observed for standard two-layer ReLU networks trained by SGD.

## 3.3 NUMERICAL VALIDATIONS

In this section, we present small-scale experiments to corroborate our theoretical results with a 4-layer linear network and two-layer ReLU network (both layers are trainable). Both isotropic and anistropic

input distributions are examined and in pariruclar, for the anistropic case, we set $\lambda_k^2(S) = 1/\sqrt{k}$. As for sample size, we set $n = 5\log(d_{\text{eff}})$ to focus on the low-sample regime. The results are reported in Figure 1 and it is evident that across all examined scenarios, the alignment strength is consistently lower-bounded and independent of the model size.

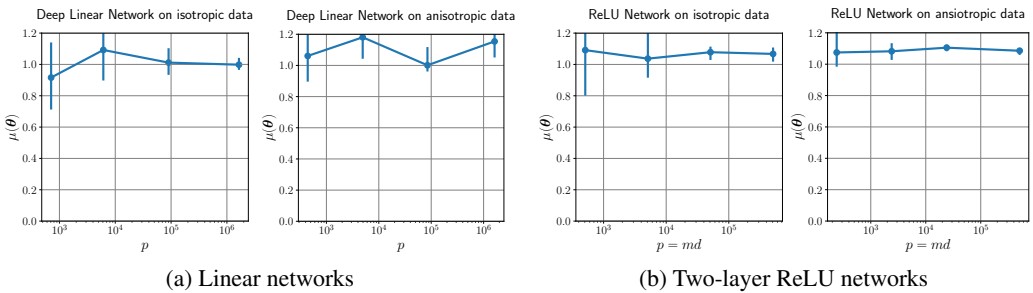

(a) Linear networks  (b) Two-layer ReLU networks

Figure 1: The alignment strength is independent of model size. Two types of models: 4-layer linear network, and two-layer neural network are examined. In experiments, we set $n = 5\log(d_{\text{eff}}), d_{\text{eff}} = 50$. The error bar corresponds to the standard deviation over 20 independent runs.

## 4   DIRECTIONAL ALIGNMENT

In Section 3, we focused solely on average alignment. Subsequently, we delve into a specific type of directional alignment: *whether noise energy along a direction is proportional to the curvature of loss landscape along that direction*. To this end, we define the following metric to measure the strength of directional alignment.

**Definition 4.1** (Directional Alignment). Given $\boldsymbol{v} \in \mathbb{R}^p$, the alignment along $\boldsymbol{v}$ is defined as

$$g(\boldsymbol{\theta}; \boldsymbol{v}) := \frac{\boldsymbol{v}^\top \Sigma(\boldsymbol{\theta}) \boldsymbol{v}}{2\mathcal{L}(\boldsymbol{\theta})\left(\boldsymbol{v}^\top G(\boldsymbol{\theta})\boldsymbol{v}\right)}, \tag{8}$$

where $\boldsymbol{v}^\top \Sigma(\boldsymbol{\theta})\boldsymbol{v} = \mathbb{E}[(\boldsymbol{\xi}(\boldsymbol{\theta})^\top \boldsymbol{v})^2]$ denotes the noise energy along direction $\boldsymbol{v}$, $\boldsymbol{v}^\top G(\boldsymbol{\theta})\boldsymbol{v}$ is the curvature of loss landscape along $\boldsymbol{v}$, and $2\mathcal{L}(\boldsymbol{\theta})$ is only a scaling factor inspired by (5).

**Theorem 4.2** (One-sided bound). *Consider OLMs and assume* $\boldsymbol{x}_1, \boldsymbol{x}_2, \ldots, \boldsymbol{x}_n \overset{\text{i.i.d.}}{\sim} \mathcal{N}(\mathbf{0}, S)$. *For any* $\delta \in (0,1)$, *if* $n \gtrsim d + \log(1/\delta)$, *then* w.p. *at least* $1 - \delta$, *we have* $\inf_{\boldsymbol{\theta}, \boldsymbol{v} \in \mathbb{R}^p} g(\boldsymbol{\theta}; \boldsymbol{v}) \gtrsim 1$.

This theorem establishes that a sample size satisfying $n \gtrsim d$ is sufficient to guarantee a uniform lower bound for alignment across all directions and the entire parameter space. The subsequent theorem builds upon this by offering a two-sided bound on alignment strength, albeit at the cost of requiring a larger sample size.

**Theorem 4.3** (Two-sided bound). *Consider OLMs and assume* $\boldsymbol{x}_1, \boldsymbol{x}_2, \ldots, \boldsymbol{x}_n \overset{\text{i.i.d.}}{\sim} \mathcal{N}(\mathbf{0}, S)$. *For any* $\epsilon, \delta \in (0,1)$, *if* $n \gtrsim \max\left\{\left(d^2 \log^2(1/\epsilon) + \log^2(1/\delta)\right)/\epsilon, (d\log(1/\epsilon) + \log(1/\delta))/\epsilon^2\right\}$, *then* w.p. *at least* $1 - \delta$, *we have the following two-side uniform bounds for the directional alignment:*

$$(i). \quad \frac{1-\epsilon}{(1+\epsilon)^2} \le \inf_{\boldsymbol{\theta}, \boldsymbol{v} \in \mathbb{R}^p} g(\boldsymbol{\theta}; \boldsymbol{v}) \le \sup_{\boldsymbol{\theta}, \boldsymbol{v} \in \mathbb{R}^p} g(\boldsymbol{\theta}; \boldsymbol{v}) \le \frac{2+\epsilon}{(1-\epsilon)^2},$$

$$(ii). \quad \frac{1-\epsilon}{(1+\epsilon)^2} \le \inf_{\boldsymbol{\theta} \in \mathbb{R}^p, \langle \boldsymbol{v}, \nabla\mathcal{L}(\boldsymbol{\theta})\rangle = 0} g(\boldsymbol{\theta}; \boldsymbol{v}) \le \sup_{\boldsymbol{\theta} \in \mathbb{R}^p, \langle \boldsymbol{v}, \nabla\mathcal{L}(\boldsymbol{\theta})\rangle = 0} g(\boldsymbol{\theta}; \boldsymbol{v}) \le \frac{1+\epsilon}{(1-\epsilon)^2}.$$

Notably, for directions satisfying $\boldsymbol{v} \perp \nabla\mathcal{L}(\boldsymbol{\theta})$, the alignment strength is nearly 1. The proofs of the above two theorems are deferred to Appendix C.

*Remark* 4.4. It is worth noting that the above theorems establish the directional alignment for all directions and the entire landscape. Consequently, the requirement of sample size is much more restricted. However, in practice, what matters are the solutions and directions explored by a certain optimizer such as SGD. This is the gap between the practice and our theory. Our experiments in Figure 2 show that indeed the directional alignment holds very well for SGD solutions and eigen-directions even when $n \ll d$. On the one hand, to formalize this insight into a theorem is challenging as it

requires a precise characterization what "SGD solutions" means. On the other hand, our theorems are also more general in the sense that it reveals that the alignment property is a intrisic property of mini-batch noise and applicable to optimizers beyond SGD.

**Numerical validations.** In this experiment, we consider the alignment along the eigen-directions of Hessian matrix. Let $G(\boldsymbol{\theta}) = \sum_k \lambda_k(\boldsymbol{\theta}) \boldsymbol{u}_k(\boldsymbol{\theta}) \boldsymbol{u}_k(\boldsymbol{\theta})^\top$ be the eigen-decomposition of $G(\boldsymbol{\theta})$ respectively, where $\{\lambda_k(\boldsymbol{\theta})\}_k$ are the eigenvalues in a decreasing order and $\{\boldsymbol{u}_k(\boldsymbol{\theta})\}$ are the corresponding eigen-directions. Note that $\lambda_k(\boldsymbol{\theta})$ is the curvature of local landscape along $\boldsymbol{u}_k(\boldsymbol{\theta})$. Decompose SGD noise along these eigen-directions: $\boldsymbol{\xi}(\boldsymbol{\theta}) = \sum_k r_k(\boldsymbol{\theta}) \boldsymbol{u}_k(\boldsymbol{\theta})$, where $r_k(\boldsymbol{\theta}) = \boldsymbol{\xi}(\boldsymbol{\theta})^\top \boldsymbol{u}_k(\boldsymbol{\theta})$ denotes the noise component in the direction of $\boldsymbol{u}_k(\boldsymbol{\theta})$. Consequently, the (scaled) expected noise magnitude in the direction $\boldsymbol{u}_k(\boldsymbol{\theta})$ is given by $\alpha_k(\boldsymbol{\theta}) = \mathbb{E}[r_k^2(\boldsymbol{\theta})]/2\mathcal{L}(\boldsymbol{\theta}) = \boldsymbol{u}_k^\top \Sigma(\boldsymbol{\theta}) \boldsymbol{u}_k(\boldsymbol{\theta})/2\mathcal{L}(\boldsymbol{\theta})$. For comparison, let $\{\mu_k(\boldsymbol{\theta})\}_k$ denote the eigenvalues of $\Sigma(\boldsymbol{\theta})/2\mathcal{L}(\boldsymbol{\theta})$. When clear from the context, we will omit dependence on $\boldsymbol{\theta}$ for simplicity.

In Figure 2a, we examine linear regression in the regimes with limited data. Surprisingly, even with significantly fewer samples, we still observed that the noise energy along each eigen-direction remained roughly proportional to the corresponding curvature and the ratio is close 1. However, we noticed that the eigenvalues of $\Sigma(\boldsymbol{\theta})/2\mathcal{L}(\boldsymbol{\theta})$ decayed much faster than that of $G(\boldsymbol{\theta})$, indicating that the condition $n \gtrsim d$ stated in Theorem 4.2 is necessary to ensure uniform alignment across all directions. In Figure 2b, we further consider the classification of CIFAR-10 with a small convolutional neural network (CNN) and fully-connected neural network (FNN). We can see that the obsevation is consistent with Figure 2a, where the alignment along eigen-directions is significant.

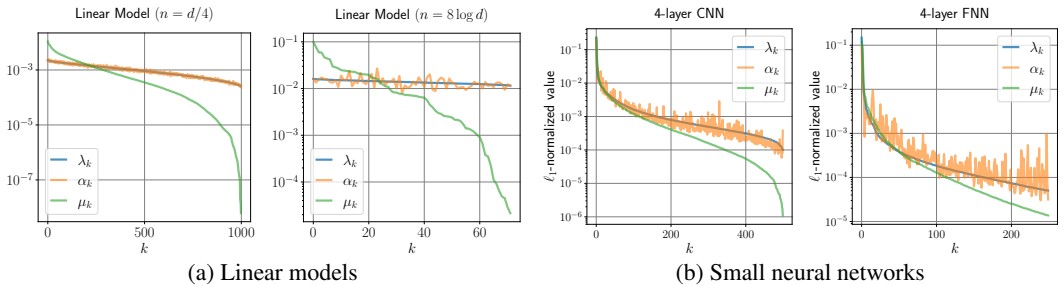

(a) Linear models        (b) Small neural networks

Figure 2: How the components of noise energy in *eigen-directions* $\{\alpha_k\}_k$ are proportional to the corresponding curvatures $\{\lambda_k\}_k$. $\alpha_k/\lambda_k$ can reflect the directional alignment (8) along the eigen-directions of the local landscape. The eigenvalues of $\Sigma/2\mathcal{L}$ are also plotted as comparison. (a) Linear models on Gaussian data in the regimes with limited data, where we fix $d = 10^3$ and change $n$ accordingly ($n = d/4$, $n = 8 \log d$). (b) 4-layer CNN and 4-layer FNN on CIFAR-10 dataset. For more experimental details, we refer to Appendix A.

## 5   HOW SGD ESCAPES FROM SHARP MINIMA

Existing analyses of the escape behavior focues on the escape rate. In this section, we provide a further analysis of the escape direction by leveraging the directional alignment. Let $\boldsymbol{\theta}^*$ be the minimum of interest. The local escape behavior can be fully characterized by linearizing the SGD dynamics, which corresponds to the linearized model $f(\cdot; \boldsymbol{\theta}) \approx f(\cdot; \boldsymbol{\theta}^*) + \langle \nabla f(\cdot; \boldsymbol{\theta}^*), \boldsymbol{\theta} - \boldsymbol{\theta}^* \rangle$. We refer to (Wu et al., 2022, Section 3.2) for more details. Thus, without loss of generality, we can simply consider the linearized model in the subsequent analysis.

Let $\boldsymbol{w} = \boldsymbol{\theta} - \boldsymbol{\theta}^*$ and $\boldsymbol{z}_i = \nabla f(\boldsymbol{x}_i; \boldsymbol{\theta}^*)$. Then, $G(\boldsymbol{\theta}^*) = \frac{1}{n} \sum_{i=1}^n \boldsymbol{z}_i \boldsymbol{z}_i^\top$ and the linearized SGD of iterates as follows

$$\boldsymbol{w}(t+1) = \boldsymbol{w}(t) - \eta \left( G(\boldsymbol{\theta}^*) \boldsymbol{w}(t) + \boldsymbol{\xi}(t) \right),$$

where $\boldsymbol{\xi}(t)$ is the SGD noise. In addition, in this section, we simply use $\mathcal{L}(\boldsymbol{w}) = \frac{1}{2} \boldsymbol{w}^T G(\boldsymbol{\theta}^*) \boldsymbol{w}$ to denote the corresponding loss. We make the following assumption on the noise alignment.

**Assumption 5.1** (Eigen-directional alignment). let $G(\boldsymbol{\theta}^*) = \sum_{i=1}^d \lambda_i \boldsymbol{u}_i \boldsymbol{u}_i^\top$ be the eigen decomposition of $G(\boldsymbol{\theta}^*)$. Assume that there exist $A_1, A_2 > 0$ such that it holds for any $\boldsymbol{w} \in \mathbb{R}^d$

$$A_1 \mathcal{L}(\boldsymbol{w}) \lambda_i \leq \mathbb{E}[|\xi(\boldsymbol{w})^\top \boldsymbol{u}_i|^2] \leq A_2 \mathcal{L}(\boldsymbol{w}) \lambda_i.$$

For linear models under the setting of Theorem 4.3, Assumption 5.1 is provably valid. It is important to clarify, however, that the above assumption only requires the alingment along eigen-directions,

which is considerably less stringent compared to the uniform directional alignment specified in Theorem 4.3. Consequently, it is plausible that Assumption 5.1 enjoys broader applicability. As empirical evidence, Figure 2b corroborates the eigen-direction alignment for fully-connected networks and CNNs when trained via SGD.

**Eigen-decomposition of SGD.** By leveraging Assumption 5.1, we can analyze the SGD dynamics in the eigenspace. Let $\boldsymbol{w}(t) = \sum_{i=1}^d w_i(t)\boldsymbol{u}_i$ with $w_i(t) = \boldsymbol{u}_i^\top \boldsymbol{w}(t)$. Then, $w_i(t+1) = (1 - \eta\lambda_i)w_i(t) + \eta\boldsymbol{\xi}(t)^\top \boldsymbol{u}_i$. Taking the expectation of the square of both sides, we obtain

$$\mathbb{E}\big[w_i^2(t+1)\big] = (1 - \eta\lambda_i)^2\mathbb{E}\big[w_i^2(t)\big] + \eta^2\mathbb{E}[|\boldsymbol{u}_i^\top \boldsymbol{\xi}(t)|^2], \qquad (9)$$

where the noise term: $\mathbb{E}[|\boldsymbol{u}_i^\top \boldsymbol{\xi}(t)|^2] \sim \lambda_i\mathcal{L}(\boldsymbol{w}_t)$ according to Assumption 5.1.

Let $X_t = \sum_{i=1}^k \lambda_i\mathbb{E}[w_i^2(t)], Y_t = \sum_{i=k+1}^d \lambda_i\mathbb{E}[w_i^2(t)]$, denoting the components of loss energy along sharp and flat directions, respectively. Let $D_k(t) = Y_t/X_t$, which measures the concentration of loss energy along flat directions. Analogously, let $P_k(t) = \sum_{i=k+1}^d \mathbb{E}[w_i^2(t)]/\sum_{i=1}^k \mathbb{E}[w_i^2(t)]$, which measure the concentration of variance along flat directions. It is easy to show that $P_k(t) \geq D_k(t)\lambda_k/\lambda_{k+1}$. Therefore, when $\lambda_k/\lambda_{k+1}$ is lower bounded, a concentration of loss energy along flat directions can lead to a similar concentration in terms of variance.

**Theorem 5.2** (Escape of SGD). *Suppose Assumption 5.1 holds and let $\eta = \frac{\beta}{\|G(\boldsymbol{\theta}^*)\|_F}$. Then, there exists absolute constants $c_1, c_2 > 0$ such that if $\beta \geq c_1$, then SGD will escape from that minima and for any $k \in [d]$, it holds that when $t \geq \max\left\{1, \frac{\log\left(c_2/\eta(\sum_{i=1}^k \lambda_i^2)^{1/2}\right)}{\log\beta}\right\}: D_k(t) \gtrsim \frac{\sum_{i=k+1}^d \lambda_i^2}{\sum_{i=1}^k \lambda_i^2}.$*

The proof can be found in Appendix D. This theorem reveals that during SGD's escape process, the loss rapidly accumulates a significant component along flat directions of the loss landscape. The precise loss ratio between the flat and sharp directions is governed by the spectrum of Hessian matrix. In particular, $D_1(t) \gtrsim \mathrm{srk}(G^2) - 1$, indicating that in high dimension, i.e., $\mathrm{srk}(G^2) \gg 1$, the loss energy along the sharpest directions becomes negligible during the SGD's escape process. This stands in stark contrast to GD, which always escapes along the sharpest direction:

**Proposition 5.3** (Escape of GD). *Consider GD with learning rate $\eta = \beta/\lambda_1$. If $\beta > 2$, then $D_1(t) \leq \sum_{i=2}^d \frac{\lambda_i(1-\eta\lambda_i)^{2t}w_i^2(0)}{\lambda_1(1-\eta\lambda_1)^{2t}w_1^2(0)}.$*

In particular, if $w_1(0) \neq 0$ and $\lambda_1 > \lambda_2$, then the above proposition implies that $D_1(t)$ decreases to 0 exponentially fast for GD.

Figure 3 presents numerical comparisons of the escaping directions between SGD and GD. It is evident that $D_1(t)$ exponentially decreases to zero for GD, indicating that GD escapes along the sharpest direction. In contrast, for SGD, $D_1(t)$ remains significantly large, indicating that SGD retains a substantial component along the flat directions during the escape process. Furthermore, the value of $D_1(t)$ positively correlates with $\mathrm{srk}(G^2)$, as predicted by our Theorem 5.2. These observations provide empirical confirmation of our theoretical predictions.

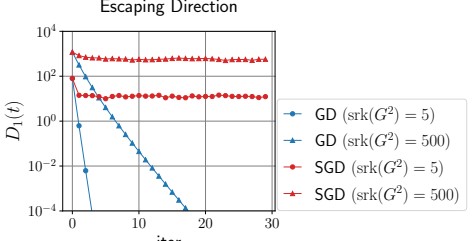

Figure 3: Comparison of escape directions between SGD and GD. The problem is linear regression and both SGD and GD are initialized near the global minimum by $\boldsymbol{w}(0) \sim \mathcal{N}(\boldsymbol{w}^*, e^{-10}I_d/d)$. To ensure escape, we choose $\eta = 1.2/\|G\|_F$ and $\eta = 4/(\lambda_1 + \lambda_2)$ for SGD and GD, respectively. Please refer to Appendix A for more experimental details.

### 5.1 EXPLAINING THE IMPLICIT BIAS OF CYCLICAL LEARNING RATE

Gaining insights into the escape direction of SGD can be valuable for understanding its optimization dynamics, generalization properties, and the overall behavior. A more detailed discussion on this topic is available in Section 7. In this section, however, we concentrate a specific example, illustrating the role of escape direction in enhancing the implicit bias of SGD through Cyclical Learning Rate (CLR) (Smith, 2017; Loshchilov and Hutter, 2017). As shown in Figure 2 of Huang et al. (2018), utilizing CLR enables SGD to cyclically escapes from (when increasing LR) and slides into (when decreasing LR) sharp regions, ultimately progressing towards flatter minima. We hypothesize that escape along flat directions plays a pivotal role in guiding SGD towards flatter region in this process.

Following Ma et al. (2022), we consider a toy OLM $f(x; \boldsymbol{w}) = (w_2/\sqrt{w_1^2 + 1})x$ with $x \sim \mathcal{N}(0, 1)$. For simplicity, we consider the online setting, where the landscape

$$\mathcal{L}(\boldsymbol{w}) = w_2^2/[2(w_1^2 + 1)].$$

The global minima valley is $S = \{\boldsymbol{w} : w_2 = 0\}$ and for $\boldsymbol{w} \in S$, $\mathrm{tr}[\nabla^2 \mathcal{L}(\boldsymbol{w})] = 1/(1 + w_1^2)$. Hence, the minimum gets flatter along the valley $S$ when $|w_1|$ grows up. In Figure 4, we visualize the trajectories for both SGD+CLR and GD+CLR. One can observe that

- SGD escape from the minima along both the flat direction $\boldsymbol{e}_1$ and sharp direction $\boldsymbol{e}_2$. The component of along $\boldsymbol{e}_1$ leads to considerable increase in $w_1^2(t)$, facilitating the movement towards flatter region along the minimum valley $S$.

- On the contrary, GD escapes only along $\boldsymbol{e}_2$, yielding no increase in $w_1^2(t)$. Thus, we cannot observe clear movement towards flatter region for GD+CLR.

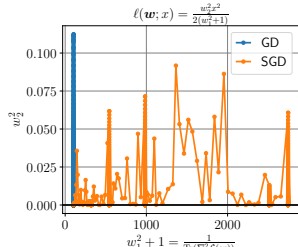

Figure 4: Visualization of the trajectories of SGD+CLR v.s. GD+CLR for our toy model. Both cases use the same CLR schedule. We can observe that SGD+CLR moves significantly towards flatter region, while GD+CLR only oscsilates along the sharpest direction. We have extensively tuned the learning rates for GD+CLR but do not obseve significant movement towards flatter region in any case.

Thus, in this toy model, the fact that SGD escapes along flat directions is crucial in amplifying the implicit bias towards flat minima.

Nonetheless, understanding how the above mechanism manifests in practice remains an open question that warrants further investigation. We defer this topic to future work, as the primary focus of this paper is to understand the noise geometry rather than exhaustively explore its applications.

## 6 LARGER-SCALE EXPERIMENTS FOR DEEP NEURAL NETWORKS

We have already provided small-scale experiments to confirm our theoretical findings. We now turn to justify the practical relevance by examining the classification of CIFAR-10 dataset (Krizhevsky and Hinton, 2009) with practical VGG nets (Simonyan and Zisserman, 2015) and ResNets (He et al., 2016). Note that larger-scale experiments on average alignment have been previously presented in Wu et al. (2022). Thus, our focus here is on investigate the directional alignment and escape direction of SGD. We refer to Appendix A for experimental details.

**The directional alignment along eigen-directions.** Figure 5 presents the directional alignments of SGD noise for ResNet-38 and VGG-13. The alignment is examined along the eigen-directions of the local landscape. The three quantities: $\lambda_k$, $\alpha_k$, and $\mu_k$ under $\ell_1$ normalization (i.e., $\lambda_k/\|\boldsymbol{\lambda}\|_1, \alpha_k/\|\boldsymbol{\alpha}\|_1, \mu_k/\|\boldsymbol{\mu}\|_1$) are plotted. Here, $\lambda_k$ and $\alpha_k$ represent the curvature and the component of noise energy along the $k$-th eigen-direction, respectively. $\mu_k$ corresponds to the $k$-th eigenvalue of the noise covariance matrix, which is included for comparison. One can see that the alignment between $\alpha_k$ and $\lambda_k$ still exists for ResNet-38 and VGG-13, but the ratio between them becomes significantly larger. As a comparison, we refer to Figure 2b, where the ratio is well-controlled for small-scale networks trained for classifying the same dataset. We hypothesize that thiis observation is consistent with our theoretical results in Section 4: one-sided bounds require much less samples.

**The escape direction of SGD.** For large models, it is computationally prohibitive to compute the quantity $D_k(t)$ since it needs to compute the whole spectrum. Thus, we consider to measure the component along different directions without reweighting. Let $\boldsymbol{\theta}^*$ be the minimum of interest and $\boldsymbol{\theta}(t)$ be SGD/GD solution at step $t$. Define $p_k(t) = \langle \boldsymbol{\theta}(t) - \boldsymbol{\theta}^*, \boldsymbol{u}_1 \rangle$ for $k = 1$ and $p_k(t) = (\sum_{i=1}^{k} \langle \boldsymbol{\theta}(t) - \boldsymbol{\theta}^*, \boldsymbol{u}_i \rangle^2)^{1/2}$ for $k > 1$; $r_k(t) = (\|\boldsymbol{\theta}(t) - \boldsymbol{\theta}^*\|^2 - p_k^2(t))^{1/2}$. Notably, $p_k(t)$ and $r_k(t)$ represent the component along sharp and flat directions, respectively.

In Figure 6, we plot $(p_k(t), r_k(t))$ for VGG-19 and ResNet-110, where we examine various $k$ values. The plots clearly demonstrate that the escape direction of SGD exhibits significant components along the flat directions. On the other hand, GD tends to escape along much sharper directions. These empirical findings align well with our theoretical findings in Section 5.

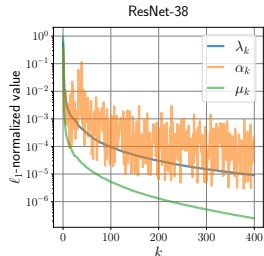
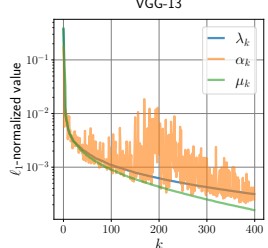

Figure 5: Three distributions ($\{\lambda_k\}_k$, $\{\alpha_k\}_k$, and $\{\mu_k\}_k$) for larger-scale neural networks, which reflect the directional alignment (8) along the eigen directions of the local landscape.

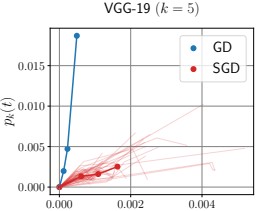
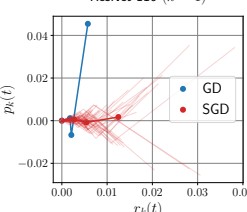
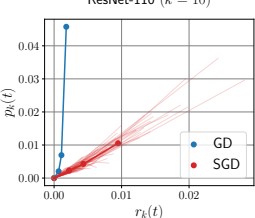

Figure 6: The red curves are 50 escaping trajectories of SGD and their average; the blue curves corresponding to GD. The sharp minimum $\theta^*$ is found by SGD. Then, we run SGD and GD starting from $\theta^*$ and the learning rates are tuned to ensure escaping.

## 7 CONCLUSION AND FUTURE WORK

In this paper, we present a comprehensive investigation of the geometry of SGD noise, demonstrating both average and directional alignment between the noise and local geometry. We substantiate these claims through both theoretical analyses and empirical evidence. Furthermore, we explore the implications of these findings by analyzing the escape direction of SGD and its role in enhancing the implicit bias toward flatter minima through cyclical learning rate.

Understanding the noise geometry is crucial for comprehending many aspects of stochastic optimization, including but not limited to convergence rates, generalization capabilities, and dynamic behavior. We offer an illustrative example through analyzing the escape direction of SGD. Another particularly relevant application of our noise geometry framework lies in deciphering the Edge of Stability (EoS) and the associated unstable convergence phenomena, as elaborated below.

- Studies (Cohen et al., 2020; Wu et al., 2018) showed that in training neural networks, GD typically occurs in a EoS phase, where the the stability condition is violated. During EoS phase, GD repeatedly slides into sharp regions and then, escapes from there. Due to the fact that GD escapes along the sharpest direction (as stated in our Proposition 5.3), GD in the EoS phase will keep *oscillating along the sharpest directions* and decreasing the loss along other flat directions. Thus, EoS facilitates the unstable convergence of GD (Ahn et al., 2022). Similar EoS-related phenomena and unstable convergence patterns are also observed in SGD (Lee and Jang, 2022). However, to fully characterize the EoS phase in the context of SGD, it is imperative to understand the underlying noise structure. Specifically, one must elucidate the mechanism by which noise compels SGD to move away from sharp minima.

- In addition, our finding can potentially be used to explain why the training curve of SGD can be more stable than that of GD—A very counter-intuitive phenomenon. As shown in Fig. 2 of Geiping et al. (2021), GD training often encounters *sudden large loss spikes* and in contrast, SGD training does not have this issue (although there are small loss fluctuations), implying that minibatch noise can stabilizes the training to some extent. This can potentially be explained by our theory as follows. For both SGD and GD, the unstable dynamics is inevitable in training neural networks due to progressive sharpening, i.e., entering the EoS phase. During the EoS phase, GD escapes along the sharpest direction, leading to a sudden large loss spike if the curvature along the sharpest direction becomes extremely large. In contrast, for SGD, the escape happens along much flatter directions, for which it is unlikely to trigger a large loss spike.

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

# Appendix

## A  EXPERIMENTAL SETUPS

In this section, we provide the experiment details for directional alignment experiments (in Figure 2 and Figure 5) and escaping experiments (in Figure 3 and Figure 6).

**Small-scale experiments** (Figure 2 and 3).

- In Figure 2, we conduct experiments on linear regression and a 4-layer linear network: $d \rightarrow m \rightarrow m \rightarrow m \rightarrow 1$ with $m = 50$. The inputs $\{\boldsymbol{x}_i\}_{i=1}^n$ are drawn from $\mathcal{N}(\mathbf{0}, I_d)$. In the first three experiments, we fix $d = 10^3$ and change $n$ accordingly ($n = 4d^2, n = d, n = d/4$). For the last experiment, we set $d = 10^4$ and $n = \log d$. Regarding the parameter $\boldsymbol{\theta}$, it is drawn from $\mathcal{N}(\mathbf{0}, I_p)$.

- In Figure 3, we conduct escaping experiments on linear regression with $\boldsymbol{w}^* = \mathbf{0}$. Both SGD and GD are initialized near the global minimum by $\boldsymbol{w}(0) \sim \mathcal{N}(\mathbf{0}, e^{-10} I_d/d)$. To ensure escaping, we choose $\eta = 1.2/ \|G\|_{\mathrm{F}}$ and $\eta = 4/(\lambda_1 + \lambda_2)$ for SGD and GD, respectively. We fix $n = 10^5$ and $d = 10^3$, and the inputs $\{\boldsymbol{x}_i\}_{i=1}^n$ are drawn from $\mathcal{N}(\mathbf{0}, \mathrm{diag}(\boldsymbol{\lambda})/d)$, where $\boldsymbol{\lambda} \in \mathbb{R}^d$ and $\lambda_1 \geq \lambda_2 = \cdots = \lambda_d \geq 0$. Moreover, we set $\lambda_1 = 1$ change $\lambda_2$ accordingly to obtain different $\mathrm{srk}(G^2)$.

**Larger-scale experiments** (Figure 5 and 6).

- Dataset. For the experiments in Figure 5 and 6, we use the CIFAR-10 dataset with label=0, 1 and the full CIFAR-10 dataset to train our models, respectively.

- Models. We conduct experiments on large-scale models: 4-layer CNN ($p = 43{,}072$), 4-layer FNN ($p = 219{,}200$), ResNet-38 ($p = 558{,}222$), VGG-13 ($p = 605{,}458$), ResNet-110 ($p = 1{,}720{,}138$), and VGG-19 ($p = 20{,}091{,}338$).

  Specifically, we use standard ResNets (He et al., 2016) and VGG nets (Simonyan and Zisserman, 2015) without batch normalization. For ResNets, we follow Zhang et al. (2019) to use the fixup initialization in order to ensure that the model can be trained without batch

normalization. Moreover, the architecture of 4-layer CNN is $\texttt{Conv}(3,6,5) \to \texttt{ReLU} \to \texttt{MPool}(2,2) \to \texttt{Conv}(6,16,5) \to \texttt{ReLU} \to \texttt{MPool}(2,2) \to \texttt{Linear}(400,100) \to \texttt{ReLU} \to \texttt{Linear}(100,2).$ and the 4-layer FNN is a ReLU-activated fully-connected network with the architecture: $784 \to 256 \to 64 \to 32 \to 2$.

- Training. All explicit regularizations (including weight decay, dropout, data augmentation, batch normalization, learning rate decay) are removed, and a simple constant-LR SGD is used to train our models. Specifically, all these models are trained by SGD with learning rate $\eta = 0.1$ and batch size $B = 32$ until the training loss becomes smaller than $10^{-4}$.

**Efficient computations** of the top-$k$ eigen-decomposition of $G$ and $\Sigma$. We utilize the functions $\texttt{eigsh}$ and $\texttt{LinearOperator}$ in $\texttt{scipy.sparse.linalg}$ to calculate top-$k$ eigenvalues and eigenvectors of $G$ and $\Sigma$, and the key step is to efficiently calculate $G\boldsymbol{v}$ and $\Sigma\boldsymbol{v}$ for any given $\boldsymbol{v} \in \mathbb{R}^p$.

- For small-scale experiments, they can be calculated directly.
- For the large-scale models, we need further approximations since the computation complexity $\mathcal{O}(np)$ is prohibitive in this case. To illustrate our method, we will use $G\boldsymbol{v}$ as an example and apply a similar approach to $\Sigma\boldsymbol{v}$. Notice that the formulation $G\boldsymbol{v} = \frac{1}{n}\sum_{i=1}^{n}(\boldsymbol{x}_i^\top \boldsymbol{v})\boldsymbol{x}_i$ are all in the form of sample average, which allows us to perform Monte-Carlo approximation. Specifically, we randomly choose $b$ samples $\{\boldsymbol{x}_{i_j}\}_{j=1}^{b}$ from $\boldsymbol{x}_1, \ldots, \boldsymbol{x}_n$ and use $\frac{1}{b}\sum_{j=1}^{b}(\boldsymbol{x}_{i_j}^\top \boldsymbol{v})\boldsymbol{x}_{i_j}$ estimate $G\boldsymbol{v}$, with the computation complexity $\mathcal{O}(bp)$. For the experiments on CIFAR-10, we test $b$'s with different values and find that $b = 2k$ is sufficient to obtain a reliable approximation of the top-$k$ eigenvalues and eigenvectors. Hence, for all large-scale experiments in this paper, we use $b = 2k$ to speed up the computation of the top-$k$ eigenvalues and eigenvectors.

# B  PROOFS IN SECTION 3: AVERAGE ALIGNMENT

## B.1  PROOF OF THEOREM 3.1 (A)

For clarity, in a slightly different order from the main text, we first prove for the linear model (Example) and then for the OLM (Theorem 3.1). This is also convenient for us to compare the difference between the proof for the two-layer neural network (Theorem 3.5) and the proof for the linear model.

**Step I.** *Proof for linear models.*

For the linear model, i.e., $\boldsymbol{\theta} = \boldsymbol{w}$ and $\boldsymbol{F}(\boldsymbol{w}) = \boldsymbol{w}$ in OLMs, we have

$$\mu(\boldsymbol{w}) = \frac{\text{Tr}\left(\Sigma(\boldsymbol{w})G(\boldsymbol{w})\right)}{2\mathcal{L}(\boldsymbol{w})\|G(\boldsymbol{w})\|_{\text{F}}^2}$$

$$= \frac{\text{Tr}\left(\left(\frac{1}{n}\sum_{j=1}^{n}\boldsymbol{x}_j\boldsymbol{x}_j^\top\right)\left(\frac{1}{n}\sum_{i=1}^{n}(F(\boldsymbol{\theta})^\top\boldsymbol{x}_i)^2(\nabla F(\boldsymbol{\theta})^\top\boldsymbol{x}_i)(\nabla F(\boldsymbol{\theta})^\top\boldsymbol{x}_i)^\top\right)\right)}{\left(\frac{1}{n}\sum_{i=1}^{n}(F(\boldsymbol{\theta})^\top\boldsymbol{x}_i)^2\right)\left(\frac{1}{n^2}\sum_{i=1}^{n}\sum_{j=1}^{n}\left(\boldsymbol{x}_i^\top\nabla F(\boldsymbol{\theta})\nabla F(\boldsymbol{\theta})^\top\boldsymbol{x}_j\right)^2\right)}$$

$$= \frac{\frac{1}{n^2}\sum_{i=1}^{n}\sum_{j=1}^{n}(\boldsymbol{w}^\top\boldsymbol{x}_i)^2\left(\boldsymbol{x}_i^\top\boldsymbol{x}_j\right)^2}{\left(\frac{1}{n}\sum_{i=1}^{n}(\boldsymbol{w}^\top\boldsymbol{x}_i)^2\right)\left(\frac{1}{n^2}\sum_{i=1}^{n}\sum_{j=1}^{n}\left(\boldsymbol{x}_i^\top\boldsymbol{x}_j\right)^2\right)} \geq \frac{\left(\frac{1}{n}\sum_{i=1}^{n}(\boldsymbol{w}^\top\boldsymbol{x}_i)^2\right)\left(\min_{i\in[n]}\frac{1}{n}\sum_{j=1}^{n}\left(\boldsymbol{x}_i^\top\boldsymbol{x}_j\right)^2\right)}{\left(\frac{1}{n}\sum_{i=1}^{n}(\boldsymbol{w}^\top\boldsymbol{x}_i)^2\right)\left(\frac{1}{n^2}\sum_{i=1}^{n}\sum_{j=1}^{n}\left(\boldsymbol{x}_i^\top\boldsymbol{x}_j\right)^2\right)} \quad (10)$$

$$= \frac{\min_{i\in[n]}\frac{1}{n}\sum_{j=1}^{n}\left(\boldsymbol{x}_i^\top\boldsymbol{x}_j\right)^2}{\max_{i\in[n]}\frac{1}{n}\sum_{j=1}^{n}\left(\boldsymbol{x}_i^\top\boldsymbol{x}_j\right)^2} \geq \frac{\min_{i\in[n]}\|\boldsymbol{x}_i\|^4 + (n-1)\min_{i\in[n]}\frac{1}{n-1}\sum_{j\neq i}(\boldsymbol{x}_i^\top\boldsymbol{x}_j)^2}{\max_{i\in[n]}\|\boldsymbol{x}_i\|^4 + (n-1)\max_{i\in[n]}\frac{1}{n-1}\sum_{j\neq i}(\boldsymbol{x}_i^\top\boldsymbol{x}_j)^2}.$$

Then we only need to estimate $\|\boldsymbol{x}_i\|^4$ and $\frac{1}{n-1}\sum_{j\neq i}(\boldsymbol{x}_i^\top\boldsymbol{x}_j)^2$ for each $i \in [n]$, respectively.

Step I (i). Estimation of $\|\boldsymbol{x}_i\|^4$.

Let $\boldsymbol{y}_i = S^{1/2}\boldsymbol{x}_i$, then $\|\boldsymbol{x}_i\|^2 = \boldsymbol{y}_i^\top S \boldsymbol{y}_i$ and $\boldsymbol{y}_1, \cdots, \boldsymbol{y}_n \overset{\text{i.i.d.}}{\sim} \mathcal{N}(\boldsymbol{0}, I_d)$.

For a fix $i \in [n]$, by Lemma E.2, there exists an absolute constant $C_1 > 0$ such that for any $\epsilon \in (0, 1)$, we have

$$\mathbb{P}\left(\left|\boldsymbol{y}_i^\top S \boldsymbol{y}_i - \text{Tr}(S)\right| \geq \epsilon \text{Tr}(S)\right) \leq 2\exp\left(-C_1 \min\left\{\frac{\epsilon^2 \text{Tr}^2(S)}{\|S\|_{\text{F}}^2}, \frac{\epsilon \text{Tr}(S)}{\|S\|_2}\right\}\right).$$

Noticing that $\text{Tr}(S)\|S\|_2 = \lambda_1 \sum_i \lambda_i \geq \sum_i \lambda_i^2 = \|S\|_F$, we thus have

$$\frac{\text{Tr}^2(S)}{\|S\|_{\text{F}}^2} \geq \frac{\text{Tr}(S)}{\|S\|_2} = \text{srk}(S).$$

Therefore,

$$\mathbb{P}\left(\left|\boldsymbol{y}_i^\top S \boldsymbol{y}_i - \text{Tr}(S)\right| \geq \epsilon \text{Tr}(S)\right) \leq 2\exp\left(-C_1 \frac{\text{Tr}(S)}{\|S\|_2} \min\left\{\epsilon, \epsilon^2\right\}\right) = 2\exp\left(-C_1\epsilon^2\text{srk}(S)\right).$$

Applying a union bound over all $i \in [n]$, we have

$$\mathbb{P}\left(\left|\|\boldsymbol{x}_i\|^2 - \text{Tr}(S)\right| \geq \epsilon \text{Tr}(S), \forall i \in [n]\right) \leq 2n\exp\left(-C_1\epsilon^2\text{srk}(S)\right).$$

In the other word, for any $\epsilon, \delta \in (0, 1)$, if $\text{srk}(S) \gtrsim \log(n)/\epsilon^2$, then *w.p.* at least $1 - \delta/3$, we have

$$(1 - \epsilon)^2 \leq \frac{\|\boldsymbol{x}_i\|_2^4}{\text{Tr}^2(S)} \leq (1 + \epsilon)^2, \ \forall i \in [n].$$

Step I (ii). Estimation of $\frac{1}{n-1}\sum_{j \neq i}(\boldsymbol{x}_i^\top \boldsymbol{x}_j)^2$.

First, we fix $i \in [n]$. Notice that $(\boldsymbol{x}_i^\top \boldsymbol{x}_j)^2$ $(j \neq i)$ are not independent, so we need estimate by some decoupling tricks.

We denote $\boldsymbol{y}_i := S^{-1/2}\boldsymbol{x}_i$, then $\boldsymbol{y}_1, \cdots, \boldsymbol{y}_n \overset{\text{i.i.d.}}{\sim} \mathcal{N}(\boldsymbol{0}, I_d)$ and $(\boldsymbol{x}_i^\top \boldsymbol{x}_j)^2 = (\boldsymbol{y}_i^\top S \boldsymbol{y}_j)^2$.

For any fixed $\boldsymbol{v} \in \mathbb{S}^{d-1}$, by Lemma E.1, for any $\epsilon \in (0, 1)$, we have

$$\mathbb{P}\left(\left|\frac{1}{n-1}\sum_{j \neq i}(\boldsymbol{v}^\top \boldsymbol{y}_j)^2 - 1\right| \geq \epsilon\right)$$

$$\leq \mathbb{P}\left(\left|\frac{1}{n-1}\sum_{j \neq i}(\boldsymbol{v}^\top \boldsymbol{y}_j)^2 - 1\right| \geq \epsilon\right) \leq 2\exp\left(-C_2(n-1)\epsilon^2\right),$$

where $C_2 > 0$ is an absolute constant, independent of $\boldsymbol{v}$ and $\epsilon$.

Then we have

$$\mathbb{P}\left(\left|\frac{1}{n-1}\sum_{j \neq i}(\boldsymbol{x}_i^\top \boldsymbol{x}_j)^2 - \boldsymbol{x}_i^\top S \boldsymbol{x}_i\right| \geq \epsilon \boldsymbol{x}_i^\top S \boldsymbol{x}_i\right)$$

$$=\mathbb{P}\left(\left|\frac{1}{n-1}\sum_{j \neq i}(\boldsymbol{y}_i^\top S \boldsymbol{y}_j)^2 - \|S\boldsymbol{y}_i\|_2^2\right| \geq \epsilon \|S\boldsymbol{y}_i\|_2^2\right)$$

$$\overset{\boldsymbol{z}_i := S\boldsymbol{y}_i/\|S\boldsymbol{y}_i\|_2}{=}\mathbb{P}\left(\left|\frac{1}{n-1}\sum_{j \neq i}(\boldsymbol{z}_i^\top \boldsymbol{y}_j)^2 - 1\right| \geq \epsilon\right)$$

$$=\mathbb{E}\left[\mathbb{I}\left\{\left|\frac{1}{n-1}\sum_{j \neq i}(\boldsymbol{z}_i^\top \boldsymbol{y}_j)^2 - 1\right| \geq 1\right\}\right]$$

$$= \mathbb{E}_{\boldsymbol{z}_i} \left[ \mathbb{E} \left[ \mathbb{I} \left\{ \left| \frac{1}{n-1} \sum_{j \neq i} (\boldsymbol{z}_i^\top \boldsymbol{y}_j)^2 - 1 \right| \geq 1 \right\} \bigg| \boldsymbol{z}_i \right] \right]$$

$$\leq \mathbb{E}_{\boldsymbol{z}_i} \left[ 2 \exp \left( -C_2 (n-1) \epsilon^2 \right) \right] = 2 \exp \left( -C_2 (n-1) \epsilon^2 \right).$$

Applying a union bound over all $i \in [n]$, we have

$$\mathbb{P} \left( \left| \frac{1}{n-1} \sum_{j \neq i} (\boldsymbol{x}_i^\top \boldsymbol{x}_j)^2 - \boldsymbol{x}_i^\top S \boldsymbol{x}_i \right| \geq \epsilon \boldsymbol{x}_i^\top S \boldsymbol{x}_i, \forall i \in [n] \right) \leq 2n \exp \left( -C_2 (n-1) \epsilon^2 \right).$$

In the other word, for any $\epsilon, \delta \in (0, 1)$, if $n / \log(n/\delta) \gtrsim 1/\epsilon^2$, then *w.p.* at least $1 - \delta/3$, we have

$$1 - \epsilon \leq \frac{\frac{1}{n-1} \sum_{j \neq i} (\boldsymbol{x}_i^\top \boldsymbol{x}_j)^2}{\boldsymbol{x}_i^\top S \boldsymbol{x}_i} \leq 1 + \epsilon, \ \forall i \in [n].$$

Step I (iii). Estimation of $\boldsymbol{x}_i^\top S \boldsymbol{x}_i$.

Let $\boldsymbol{y}_i = S^{1/2} \boldsymbol{x}_i$, then $\boldsymbol{x}_i^\top S \boldsymbol{x}_i = \boldsymbol{y}_i^\top S^2 \boldsymbol{y}_i$ and $\boldsymbol{y}_1, \cdots, \boldsymbol{y}_n \overset{\text{i.i.d.}}{\sim} \mathcal{N}(\boldsymbol{0}, I_d)$.

In the same way as Step I(i), we obtain that: for any $\epsilon, \delta \in (0, 1)$, if $\text{srk}(S^2) \gtrsim \log(n)/\epsilon^2$, then *w.p.* at least $1 - \delta/3$, we have

$$1 - \epsilon \leq \frac{\boldsymbol{x}_i^\top S \boldsymbol{x}_i}{\text{Tr}(S^2)} \leq 1 + \epsilon, \ \forall i \in [n].$$

Combining our results in Step I (i), Step I (ii), and Step I (iii), we obtain the result for Linear Model: for any $\epsilon, \delta \in (0, 1)$, if $n / \log(n/\delta) \gtrsim 1/\epsilon^2$ and $\min\{\text{srk}(S), \text{srk}(S^2)\} \gtrsim \log(n)/\epsilon^2$, then *w.p.* at least $1 - \delta/3 - \delta/3 - \delta/3 = 1 - \delta$, we have

$$\mu(\boldsymbol{w}) \geq \frac{(1-\epsilon)^2 \text{Tr}^2(S) + (n-1)(1-\epsilon) \min_{i \in [n]} \boldsymbol{x}_i^\top S \boldsymbol{x}_i}{(1+\epsilon)^2 \text{Tr}^2(S) + (n-1)(1+\epsilon) \max_{i \in [n]} \boldsymbol{x}_i^\top S \boldsymbol{x}_i}$$

$$\geq \frac{(1-\epsilon)^2 \text{Tr}^2(S) + (n-1)(1-\epsilon)^2 \text{Tr}(S^2)}{(1+\epsilon)^2 \text{Tr}^2(S) + (n-1)(1+\epsilon)^2 \text{Tr}(S^2)} = \frac{(1-\epsilon)^2}{(1+\epsilon)^2}.$$

From the arbitrary of $\boldsymbol{w}$, we have $\inf_{\boldsymbol{w} \in \mathbb{R}^d} \mu(\boldsymbol{w}) \geq \frac{(1-\epsilon)^2}{(1+\epsilon)^2}$.

**Step II.** *Proof for OLMs.*

$$\mu(\boldsymbol{\theta}) = \frac{\mathrm{Tr}\left(\Sigma(\boldsymbol{\theta})G(\boldsymbol{\theta})\right)}{2\mathcal{L}(\boldsymbol{\theta})\left\|G(\boldsymbol{\theta})\right\|_{\mathrm{F}}^2}$$

$$= \frac{\mathrm{Tr}\left(\left(\frac{1}{n}\sum_{j=1}^n (\nabla F(\boldsymbol{\theta})^\top \boldsymbol{x}_j)(\nabla F(\boldsymbol{\theta})^\top \boldsymbol{x}_j)^\top\right)\left(\frac{1}{n}\sum_{i=1}^n (F(\boldsymbol{\theta})^\top \boldsymbol{x}_i)^2(\nabla F(\boldsymbol{\theta})^\top \boldsymbol{x}_i)(\nabla F(\boldsymbol{\theta})^\top \boldsymbol{x}_i)^\top\right)\right)}{\left(\frac{1}{n}\sum_{i=1}^n (F(\boldsymbol{\theta})^\top \boldsymbol{x}_i)^2\right)\left(\frac{1}{n^2}\sum_{i=1}^n\sum_{j=1}^n \left(\boldsymbol{x}_i^\top \nabla F(\boldsymbol{\theta})\nabla F(\boldsymbol{\theta})^\top \boldsymbol{x}_j\right)^2\right)}$$

$$= \frac{\frac{1}{n^2}\sum_{i=1}^n\sum_{j=1}^n (F(\boldsymbol{\theta})^\top \boldsymbol{x}_i)^2\left(\boldsymbol{x}_i^\top \nabla F(\boldsymbol{\theta})\nabla F(\boldsymbol{\theta})^\top \boldsymbol{x}_j\right)^2}{\left(\frac{1}{n}\sum_{i=1}^n (F(\boldsymbol{\theta})^\top \boldsymbol{x}_i)^2\right)\left(\frac{1}{n^2}\sum_{i=1}^n\sum_{j=1}^n \left(\boldsymbol{x}_i^\top \nabla F(\boldsymbol{\theta})\nabla F(\boldsymbol{\theta})^\top \boldsymbol{x}_j\right)^2\right)}$$

$$\geq \frac{\left(\frac{1}{n}\sum_{i=1}^n (F(\boldsymbol{\theta})^\top \boldsymbol{x}_i)^2\right)\left(\min_{i\in[n]}\frac{1}{n}\sum_{j=1}^n \left(\boldsymbol{x}_i^\top \nabla F(\boldsymbol{\theta})\nabla F(\boldsymbol{\theta})^\top \boldsymbol{x}_j\right)^2\right)}{\left(\frac{1}{n}\sum_{i=1}^n (F(\boldsymbol{\theta})^\top \boldsymbol{x}_i)^2\right)\left(\frac{1}{n^2}\sum_{i=1}^n\sum_{j=1}^n \left(\boldsymbol{x}_i^\top \nabla F(\boldsymbol{\theta})\nabla F(\boldsymbol{\theta})^\top \boldsymbol{x}_j\right)^2\right)} = \frac{\min_{i\in[n]}\frac{1}{n}\sum_{j=1}^n \left(\boldsymbol{x}_i^\top \nabla F(\boldsymbol{\theta})\nabla F(\boldsymbol{\theta})^\top \boldsymbol{x}_j\right)^2}{\max_{i\in[n]}\frac{1}{n}\sum_{j=1}^n \left(\boldsymbol{x}_i^\top \nabla F(\boldsymbol{\theta})\nabla F(\boldsymbol{\theta})^\top \boldsymbol{x}_j\right)^2}$$

$$\geq \frac{\min_{i\in[n]}\left\|\nabla F(\boldsymbol{\theta})^\top \boldsymbol{x}_i\right\|^4 + (n-1)\min_{i\in[n]}\frac{1}{n-1}\sum_{j\neq i}(\boldsymbol{x}_i^\top \nabla F(\boldsymbol{\theta})\nabla F(\boldsymbol{\theta})^\top \boldsymbol{x}_j)^2}{\max_{i\in[n]}\left\|\nabla F(\boldsymbol{\theta})^\top \boldsymbol{x}_i\right\|^4 + (n-1)\max_{i\in[n]}\frac{1}{n-1}\sum_{j\neq i}(\boldsymbol{x}_i^\top \nabla F(\boldsymbol{\theta})\nabla F(\boldsymbol{\theta})^\top \boldsymbol{x}_j)^2}.$$

$$(11)$$

We can still prove the theorem by the similar way as Step I.

By replacing $\boldsymbol{x}_i$ and $\boldsymbol{x}_j$ ($j \neq i$) in Step I (i) with $\nabla F(\boldsymbol{\theta})\nabla F(\boldsymbol{\theta})^\top \boldsymbol{x}_i$ and $\boldsymbol{x}_j$ ($j \neq i$), respectively, in the similar way as Step I (i), we can obtain: for any $\epsilon, \delta \in (0,1)$, if $n/\log(n/\delta) \gtrsim 1/\epsilon^2$, then *w.p.* at least $1 - \delta$, we have

$$1 - \epsilon \leq \frac{\frac{1}{n-1}\sum_{j\neq i}(\boldsymbol{x}_i^\top \nabla F(\boldsymbol{\theta})\nabla F(\boldsymbol{\theta})^\top \boldsymbol{x}_j)^2}{\boldsymbol{x}_i^\top \nabla F(\boldsymbol{\theta})\nabla F(\boldsymbol{\theta})^\top S\nabla F(\boldsymbol{\theta})\nabla F(\boldsymbol{\theta})^\top \boldsymbol{x}_i} \leq 1 + \epsilon, \ \forall i \in [n];$$

Combining the estimation above with Step I (ii) and Step I (iii), we obtain that: for any $\epsilon, \delta \in (0,1)$, if $n/\log(n/\delta) \gtrsim 1/\epsilon^2$ and $\mathrm{srk}(S^2) \gtrsim \log(n)/\epsilon^2$, then *w.p.* at least $1 - \delta$, we have

$$1 - \epsilon \leq \frac{\frac{1}{n-1}\sum_{j\neq i}(\boldsymbol{x}_i^\top \nabla F(\boldsymbol{\theta})\nabla F(\boldsymbol{\theta})^\top \boldsymbol{x}_j)^2}{\boldsymbol{x}_i^\top \nabla F(\boldsymbol{\theta})\nabla F(\boldsymbol{\theta})^\top S\nabla F(\boldsymbol{\theta})\nabla F(\boldsymbol{\theta})^\top \boldsymbol{x}_i} \leq 1 + \epsilon, \ \forall i \in [n];$$

$$(1 - \epsilon)^2 \leq \frac{\|\boldsymbol{x}_i\|_2^4}{\mathrm{Tr}^2(S)} \leq (1 + \epsilon)^2, \ \forall i \in [n];$$

$$1 - \epsilon \leq \frac{\boldsymbol{x}_i^\top S\boldsymbol{x}_i}{\mathrm{Tr}(S^2)} \leq 1 + \epsilon, \ \forall i \in [n].$$

These inequalities imply that:

$$\mu(\boldsymbol{\theta}) \geq \frac{\min_{i\in[n]}\lambda_{\min}^2(\nabla F(\boldsymbol{\theta})\nabla F(\boldsymbol{\theta})^\top)\|\boldsymbol{x}_i\|_2^4 + (n-1)\min_{i\in[n]}\frac{1}{n-1}\sum_{j\neq i}(\boldsymbol{x}_i^\top \nabla F(\boldsymbol{\theta})\nabla F(\boldsymbol{\theta})^\top \boldsymbol{x}_j)^2}{\max_{i\in[n]}\lambda_{\min}^2(\nabla F(\boldsymbol{\theta})\nabla F(\boldsymbol{\theta})^\top)\|\boldsymbol{x}_i\|_2^4 + (n-1)\max_{i\in[n]}\frac{1}{n-1}\sum_{j\neq i}(\boldsymbol{x}_i^\top \nabla F(\boldsymbol{\theta})\nabla F(\boldsymbol{\theta})^\top \boldsymbol{x}_j)^2}$$

$$\geq \frac{(1-\epsilon)^2 \min_{i\in[n]}\lambda_{\min}^2(\nabla F(\boldsymbol{\theta})\nabla F(\boldsymbol{\theta})^\top)\mathrm{Tr}^2(S) + (n-1)(1-\epsilon)\min_{i\in[n]}\boldsymbol{x}_i^\top \nabla F(\boldsymbol{\theta})\nabla F(\boldsymbol{\theta})^\top S\nabla F(\boldsymbol{\theta})\nabla F(\boldsymbol{\theta})^\top \boldsymbol{x}_i}{(1-\epsilon)^2 \max_{i\in[n]}\lambda_{\max}^2(\nabla F(\boldsymbol{\theta})\nabla F(\boldsymbol{\theta})^\top)\mathrm{Tr}^2(S) + (n-1)(1+\epsilon)\max_{i\in[n]}\boldsymbol{x}_i^\top \nabla F(\boldsymbol{\theta})\nabla F(\boldsymbol{\theta})^\top S\nabla F(\boldsymbol{\theta})\nabla F(\boldsymbol{\theta})^\top \boldsymbol{x}_i}$$

$$\geq \frac{(1-\epsilon)^2 \min_{i\in[n]}\lambda_{\min}^2(\nabla F(\boldsymbol{\theta})\nabla F(\boldsymbol{\theta})^\top)\mathrm{Tr}^2(S) + (n-1)(1-\epsilon)\lambda_{\min}^2(\nabla F(\boldsymbol{\theta})\nabla F(\boldsymbol{\theta})^\top)\min_{i\in[n]}\boldsymbol{x}_i^\top S\boldsymbol{x}_i}{(1+\epsilon)^2 \max_{i\in[n]}\lambda_{\max}^2(\nabla F(\boldsymbol{\theta})\nabla F(\boldsymbol{\theta})^\top)\mathrm{Tr}^2(S) + (n-1)(1+\epsilon)\lambda_{\max}^2(\nabla F(\boldsymbol{\theta})\nabla F(\boldsymbol{\theta})^\top)\max_{i\in[n]}\boldsymbol{x}_i^\top S\boldsymbol{x}_i}$$

$$\geq \frac{(1-\epsilon)^2 \min_{i\in[n]} \lambda_{\min}^2(\nabla F(\boldsymbol{\theta})\nabla F(\boldsymbol{\theta})^\top) \operatorname{Tr}^2(S) + (n-1)(1-\epsilon)^2\lambda_{\min}^2(\nabla F(\boldsymbol{\theta})\nabla F(\boldsymbol{\theta})^\top) \operatorname{Tr}(S^2)}{(1+\epsilon)^2 \max_{i\in[n]} \lambda_{\max}^2(\nabla F(\boldsymbol{\theta})\nabla F(\boldsymbol{\theta})^\top) \operatorname{Tr}^2(S) + (n-1)(1+\epsilon)^2\lambda_{\max}^2(\nabla F(\boldsymbol{\theta})\nabla F(\boldsymbol{\theta})^\top) \operatorname{Tr}(S^2)}$$

$$= \frac{(1-\epsilon)^2}{(1+\epsilon)^2 \operatorname{cond}^2(\nabla F(\boldsymbol{\theta})\nabla F(\boldsymbol{\theta})^\top)}.$$

Hence, we have proved Theorem 3.1. □

## B.2 Proof of Theorem 3.1 (b)

This result is a direct corollary of Theorem 4.2, which is proved in Appendix C.

Under the same setting as Theorem 4.2, Theorem 4.2 gives us the uniform lower bound: there exists an absolute constant $C > 0$ such that

$$\inf_{\boldsymbol{\theta},\boldsymbol{v}\in\mathbb{R}^p} g(\boldsymbol{\theta};\boldsymbol{v}) \geq C,$$

which means that for any $\boldsymbol{\theta} \in \mathbb{R}^p, \boldsymbol{v} \in \mathbb{S}^{p-1}$, we have

$$\boldsymbol{v}^\top \Sigma(\boldsymbol{\theta})\boldsymbol{v} \geq C \cdot 2\mathcal{L}(\boldsymbol{\theta})\boldsymbol{v}^\top G(\boldsymbol{\theta})\boldsymbol{v}.$$

Consider the orthogonal decomposition of $G(\boldsymbol{\theta})$: $G(\boldsymbol{\theta}) = \sum_{k=1}^p \lambda_k \boldsymbol{u}_k \boldsymbol{u}_k^\top$. Notice that

$$\operatorname{Tr}(\Sigma(\boldsymbol{\theta})G(\boldsymbol{\theta})) = \sum_{k=1}^p \lambda_k \boldsymbol{u}_k^\top \Sigma(\boldsymbol{\theta})\boldsymbol{u}_k,$$

$$\|G(\boldsymbol{\theta})\|_{\mathrm{F}} = \operatorname{Tr}(G(\boldsymbol{\theta})G(\boldsymbol{\theta})) = \sum_{k=1}^p \lambda_k \boldsymbol{u}_k^\top G(\boldsymbol{\theta})\boldsymbol{u}_k.$$

Then we obtain

$$\operatorname{Tr}(\Sigma(\boldsymbol{\theta})G(\boldsymbol{\theta})) \geq C \cdot 2\mathcal{L}(\boldsymbol{\theta}) \sum_{k=1}^p \lambda_k \boldsymbol{u}_k^\top G(\boldsymbol{\theta})\boldsymbol{u}_k = C \cdot 2\mathcal{L}(\boldsymbol{\theta}) \|G(\boldsymbol{\theta})\|_{\mathrm{F}}^2,$$

which means $\mu(\boldsymbol{\theta}) \geq C$. From the arbitrariness of $\boldsymbol{\theta}$, it holds that $\inf_{\boldsymbol{\theta}\in\mathbb{R}^p} \mu(\boldsymbol{\theta}) \geq C$. □

## B.3 Proof of Theorem 3.5

For two-layer neural networks with fixed output layer, the gradient is

$$\nabla f(\boldsymbol{x}_i;\boldsymbol{\theta}) = \left(a_1\sigma'(\boldsymbol{b}_1^\top \boldsymbol{x}_i)\boldsymbol{x}_i^\top, \cdots, a_m\sigma'(\boldsymbol{b}_m^\top \boldsymbol{x}_i)\boldsymbol{x}_i^\top\right)^\top \in \mathbb{R}^{md}.$$

For simplicity, denote $\nabla f_i(\boldsymbol{\theta}) := \nabla f(\boldsymbol{x}_i;\boldsymbol{\theta})$, $\boldsymbol{u}_i(\boldsymbol{\theta}) := f_i(\boldsymbol{\theta}) - f_i(\boldsymbol{\theta}^*)$. Then we have:

$$\mathcal{L}(\boldsymbol{\theta}) = \frac{1}{2n}\sum_{i=1}^n u_i^2(\boldsymbol{\theta}), \quad G(\boldsymbol{\theta}) = \frac{1}{n}\sum_{i=1}^n \nabla f_i(\boldsymbol{\theta})\nabla f_i(\boldsymbol{\theta})^\top, \quad \Sigma(\boldsymbol{\theta}) = \frac{1}{n}\sum_{i=1}^n u_i^2(\boldsymbol{\theta})\nabla f_i(\boldsymbol{\theta})\nabla f_i(\boldsymbol{\theta})^\top.$$

$$\mu(\boldsymbol{\theta}) = \frac{\operatorname{Tr}\left(\left(\frac{1}{n}\sum_{i=1}^n \nabla f_i(\boldsymbol{\theta})\nabla f_i(\boldsymbol{\theta})^\top\right)\left(\frac{1}{n}\sum_{i=1}^n u_i^2(\boldsymbol{\theta})\nabla f_i(\boldsymbol{\theta})\nabla f_i(\boldsymbol{\theta})^\top\right)\right)}{\left(\frac{1}{n}\sum_{i=1}^n u_i^2(\boldsymbol{\theta})\right)\left(\frac{1}{n^2}\sum_{i=1}^n\sum_{j=1}^n \left(\nabla f_i(\boldsymbol{\theta})^\top \nabla f_i(\boldsymbol{\theta})\right)^2\right)}$$

$$= \frac{\frac{1}{n}\sum_{i=1}^n u_i^2(\boldsymbol{\theta})\frac{1}{n}\sum_{j=1}^n \left(\nabla f_i(\boldsymbol{\theta})^\top \nabla f_j(\boldsymbol{\theta})\right)^2}{\left(\frac{1}{n}\sum_{i=1}^n u_i^2(\boldsymbol{\theta})\right)\left(\frac{1}{n^2}\sum_{i=1}^n\sum_{j=1}^n \left(\nabla f_i(\boldsymbol{\theta})^\top \nabla f_i(\boldsymbol{\theta})\right)^2\right)}$$

$$\geq \frac{\min\limits_{i\in[n]} \frac{1}{n}\sum\limits_{j=1}^{n}\left(\nabla f_i(\boldsymbol{\theta})^\top \nabla f_j(\boldsymbol{\theta})\right)^2}{\frac{1}{n^2}\sum\limits_{i=1}^{n}\sum\limits_{i=1}^{n}\left(\nabla f_i(\boldsymbol{\theta})^\top \nabla f_j(\boldsymbol{\theta})\right)^2} \geq \frac{\min\limits_{i\in[n]} \frac{1}{n}\sum\limits_{j=1}^{n}\left(\alpha^2 m\boldsymbol{x}_i^\top \boldsymbol{x}_j\right)^2}{\frac{1}{n^2}\sum\limits_{i=1}^{n}\sum\limits_{i=1}^{n}\left(\beta^2 m\boldsymbol{x}_i^\top \boldsymbol{x}_j\right)^2} = \frac{\alpha^2}{\beta^2}\frac{\min\limits_{i\in[n]} \frac{1}{n}\sum\limits_{j=1}^{n}\left(\boldsymbol{x}_i^\top \boldsymbol{x}_j\right)^2}{\frac{1}{n^2}\sum\limits_{i=1}^{n}\sum\limits_{i=1}^{n}\left(\boldsymbol{x}_i^\top \boldsymbol{x}_j\right)^2}.$$

Notice that the last term $\dfrac{\min\limits_{i\in[n]} \frac{1}{n}\sum\limits_{j=1}^{n}\left(\boldsymbol{x}_i^\top \boldsymbol{x}_j\right)^2}{\frac{1}{n^2}\sum\limits_{i=1}^{n}\sum\limits_{i=1}^{n}\left(\boldsymbol{x}_i^\top \boldsymbol{x}_j\right)^2}$ is independent of $\boldsymbol{\theta}$ and the same as (10) for the linear model . Then repeating the same proof of Linear Model, the result of this theorem differs from Linear Model by only the factor $\alpha^2/\beta^2$. In other words, under the same condition with Linear Model, *w.p.* at least $1 - \delta$, we have

$$\inf_{\boldsymbol{\theta}\in\mathbb{R}^{md}} \mu(\boldsymbol{\theta}) \geq \frac{\alpha^2}{\beta^2}\frac{(1-\epsilon)^2}{(1+\epsilon)^2}.$$

$\square$

## C  PROOFS IN SECTION 4: DIRECTIONAL ALIGNMENT

For the OLM $f(\boldsymbol{x}; \boldsymbol{\theta}) = F(\boldsymbol{\theta})^T \boldsymbol{x}$, let $\boldsymbol{r}(\boldsymbol{\theta}) = F(\boldsymbol{\theta}) - F(\boldsymbol{\theta}^*)$. Then, we have

$$\hat{G}(\boldsymbol{\theta}) = \frac{1}{n}\sum_{i=1}^{n}\nabla F^\top(\boldsymbol{\theta})\boldsymbol{x}_i\boldsymbol{x}_i^\top \nabla F(\boldsymbol{\theta})$$

$$\hat{\mathcal{L}}(\boldsymbol{\theta}) = \frac{1}{2n}\sum_{i=1}^{n}\left(\boldsymbol{u}^\top(\boldsymbol{\theta})\boldsymbol{x}_i\right)^2 \tag{12}$$

$$\hat{\Sigma}(\boldsymbol{\theta}) = \frac{1}{n}\sum_{i=1}^{n}\left(\boldsymbol{r}^\top(\boldsymbol{\theta})\boldsymbol{x}_i\right)^2 \nabla F^\top(\boldsymbol{\theta})\boldsymbol{x}_i\boldsymbol{x}_i^\top \nabla F(\boldsymbol{\theta}),$$

and for the population case:

$$G(\boldsymbol{\theta}) = \mathbb{E}\left[\nabla F^\top(\boldsymbol{\theta})\boldsymbol{x}\boldsymbol{x}^\top \nabla F(\boldsymbol{\theta})\right] = \nabla F^\top(\boldsymbol{\theta})S\nabla F(\boldsymbol{\theta})$$

$$\mathcal{L}(\boldsymbol{\theta}) = \frac{1}{2}\mathbb{E}\left[\left(\boldsymbol{r}^\top(\boldsymbol{\theta})\boldsymbol{x}\right)^2\right] = \frac{1}{2}\boldsymbol{r}(\boldsymbol{\theta})^\top S\boldsymbol{r}(\boldsymbol{\theta})$$

$$\Sigma(\boldsymbol{\theta}) = \mathbb{E}\left[\left(\boldsymbol{r}^\top(\boldsymbol{\theta})\boldsymbol{x}\right)^2 \nabla F^\top(\boldsymbol{\theta})\boldsymbol{x}\boldsymbol{x}^\top \nabla F(\boldsymbol{\theta})\right]$$

**Lemma C.1** (Proposition 2.3 in (Wu et al., 2022)). *Let the data distribution be* $\mathcal{N}(\mathbf{0}, S)$. *Then we have*
$$\Sigma(\boldsymbol{\theta}) = \nabla\mathcal{L}(\boldsymbol{\theta})\nabla\mathcal{L}(\boldsymbol{\theta})^\top + 2\mathcal{L}(\boldsymbol{\theta})G(\boldsymbol{\theta}).$$

**Lemma C.2.** *Under the same conditions in Lemma C.1, if* $\boldsymbol{u}(\boldsymbol{\theta}) \neq \mathbf{0}$ *and* $\nabla F(\boldsymbol{\theta})\boldsymbol{v} \neq 0$, *then we have:*
$$\left(\nabla\mathcal{L}(\boldsymbol{\theta})^\top \boldsymbol{v}\right)^2 \leq 2\mathcal{L}(\boldsymbol{\theta})\boldsymbol{v}^\top G(\boldsymbol{\theta})\boldsymbol{v}.$$

*Proof.* Noticing that $\mathcal{L}(\boldsymbol{\theta}) = \frac{1}{2}\boldsymbol{r}(\boldsymbol{\theta})^\top S\boldsymbol{r}(\boldsymbol{\theta})$, we have $\nabla\mathcal{L}(\boldsymbol{\theta}) = \nabla F(\boldsymbol{\theta})^\top S\boldsymbol{u}(\boldsymbol{\theta})$. Hence,

$$\left(\nabla\mathcal{L}(\boldsymbol{\theta})^\top \boldsymbol{v}\right)^2 = \boldsymbol{v}^\top \nabla F(\boldsymbol{\theta})^\top S\boldsymbol{r}(\boldsymbol{\theta})\boldsymbol{r}(\boldsymbol{\theta})^\top S\nabla F(\boldsymbol{\theta})\boldsymbol{v} = \langle\nabla F(\boldsymbol{\theta})\boldsymbol{v}, \boldsymbol{r}(\boldsymbol{\theta})\rangle_S^2$$

$$\overset{\text{Lemma E.6}}{\leq} \|\nabla F(\boldsymbol{\theta})\boldsymbol{v}\|_S^2 \|\boldsymbol{r}(\boldsymbol{\theta})\|_S^2 = 2\mathcal{L}(\boldsymbol{\theta})\left(\boldsymbol{v}\nabla F(\boldsymbol{\theta})^\top S\nabla F(\boldsymbol{\theta})\boldsymbol{v}\right) = 2\mathcal{L}(\boldsymbol{\theta})\boldsymbol{v}^\top G(\boldsymbol{\theta})\boldsymbol{v}.$$

$\square$

**Lemma C.3.** *Let* $\boldsymbol{x}_1, \cdots, \boldsymbol{x}_n \overset{\text{i.i.d.}}{\sim} \mathcal{N}(\mathbf{0}, \boldsymbol{I}_d)$. *For any* $\epsilon, \delta \in (0, 1)$, *if we choose* $n \gtrsim (d + \log(1/\delta))/\epsilon^2$, *then w.p. at least* $1 - \delta$, *we have:*

$$\sup_{\boldsymbol{v}\in\mathbb{S}^{d-1}}\left|\frac{1}{n}\sum_{i=1}^{n}(\boldsymbol{v}^\top \boldsymbol{x}_i)^2 - 1\right| \leq \epsilon.$$

*Proof.* By Lemma E.3 with $K = \sqrt{C_1}$, we know that: *w.p.* at least $1 - 2\exp(-u)$, we have

$$\left\|\frac{1}{n}\sum_{i=1}^{n}\boldsymbol{x}_i\boldsymbol{x}_i^\top - \boldsymbol{I}_d\right\| \le C_2\left(\sqrt{\frac{d+u}{n}} + \frac{d+u}{n}\right),$$

where $C_2$ is an absolute positive constant. Equivalently, we can rewrite this conclusion. For any $\epsilon, \delta \in (0, 1)$, if we choose $n \gtrsim (d + \log(1/\delta))/\epsilon^2$, then *w.p.* at least $1 - \delta$, we have:

$$\sup_{\boldsymbol{v}\in\mathbb{S}^{d-1}}\left|\frac{1}{n}\sum_{i=1}^{n}(\boldsymbol{v}^\top\boldsymbol{x}_i)^2 - 1\right| \le \left\|\frac{1}{n}\sum_{i=1}^{n}\boldsymbol{x}_i\boldsymbol{x}_i^\top - \boldsymbol{I}_d\right\| \le \epsilon.$$

$\square$

**Lemma C.4** (Corollary 2 in (Cai et al., 2022)). *Let $\boldsymbol{x}_1, \cdots, \boldsymbol{x}_n \overset{\text{i.i.d.}}{\sim} \mathcal{N}(\boldsymbol{0}, \boldsymbol{I}_d)$. There exists absolute constants $C_1, C_2, C_3 > 0$, such that if $n \ge C_3 d$, then* w.p. *at least $1 - \exp(-C_2 n)$, we have*

$$\inf_{\boldsymbol{u},\boldsymbol{v}\in\mathbb{S}^{d-1}}\frac{1}{n}\sum_{i=1}^{n}(\boldsymbol{x}_i^\top\boldsymbol{u})^2(\boldsymbol{x}_i^\top\boldsymbol{v})^2 \ge C_1.$$

With the preparation of Lemma C.3 and Lemma C.4, now we give the proof of Theorem 4.2.

### C.1 PROOF OF THEOREM 4.2

Let $\boldsymbol{y}_i = \boldsymbol{S}^{-1/2}\boldsymbol{x}_i$, then $\boldsymbol{y}_1, \cdots, \boldsymbol{y}_n \overset{\text{i.i.d.}}{\sim} \mathcal{N}(\boldsymbol{0}, I_d)$.

$$
\begin{aligned}
g(\boldsymbol{\theta}; \boldsymbol{v}) &= = \frac{\frac{1}{n}\sum_{i=1}^{n}\left(\boldsymbol{r}^\top(\boldsymbol{\theta})\boldsymbol{x}_i\right)^2\left((\nabla F(\boldsymbol{\theta})\boldsymbol{v})^\top\boldsymbol{x}_i\right)^2}{\frac{1}{n}\sum_{i=1}^{n}\left(\boldsymbol{r}^\top(\boldsymbol{\theta})\boldsymbol{x}_i\right)^2 \cdot \frac{1}{n}\sum_{i=1}^{n}\left((\nabla F(\boldsymbol{\theta})\boldsymbol{v})^\top\boldsymbol{x}_i\right)^2} \\
&= \frac{\frac{1}{n}\sum_{i=1}^{n}\left((\boldsymbol{S}^{1/2}\boldsymbol{r}(\boldsymbol{\theta}))^\top\boldsymbol{y}_i\right)^2\left((\boldsymbol{S}^{1/2}\nabla F(\boldsymbol{\theta})\boldsymbol{v})^\top\boldsymbol{y}_i\right)^2}{\frac{1}{n}\sum_{i=1}^{n}\left((\boldsymbol{S}^{1/2}\boldsymbol{r}(\boldsymbol{\theta}))^\top\boldsymbol{y}_i\right)^2 \cdot \frac{1}{n}\sum_{i=1}^{n}\left((\boldsymbol{S}^{1/2}\nabla F(\boldsymbol{\theta})\boldsymbol{v})^\top\boldsymbol{y}_i\right)^2},
\end{aligned}
$$

Case(i). If $\boldsymbol{S}^{1/2}\boldsymbol{r}(\boldsymbol{\theta}) = \boldsymbol{0}$ or $\boldsymbol{S}^{1/2}\nabla F(\boldsymbol{\theta})\boldsymbol{v} = 0$, we have $g(\boldsymbol{\theta}; \boldsymbol{v}) = \frac{0}{0} = 1$, this theorem holds.

Case (ii). If $\boldsymbol{S}^{1/2}\boldsymbol{r}(\boldsymbol{\theta}) \ne \boldsymbol{0}$ and $\boldsymbol{S}^{1/2}\nabla F(\boldsymbol{\theta})\boldsymbol{v} \ne 0$, we define the following normalized vectors:

$$\tilde{\boldsymbol{r}}(\boldsymbol{\theta}) := \frac{\boldsymbol{S}^{1/2}\boldsymbol{r}(\boldsymbol{\theta})}{\left\|\boldsymbol{S}^{1/2}\boldsymbol{r}(\boldsymbol{\theta})\right\|} \in \mathbb{S}^{d-1} \quad \tilde{\boldsymbol{w}}(\boldsymbol{\theta}; \boldsymbol{v}) := \frac{\boldsymbol{S}^{1/2}\nabla F(\boldsymbol{\theta})\boldsymbol{v}}{\left\|\boldsymbol{S}^{1/2}\nabla F(\boldsymbol{\theta})\boldsymbol{v}\right\|} \in \mathbb{S}^{d-1}.$$

From the homogeneity of $g(\boldsymbol{\theta}; \boldsymbol{v})$, we have:

$$g(\boldsymbol{\theta}; \boldsymbol{v}) = \frac{\frac{1}{n}\sum_{i=1}^{n}\left(\tilde{\boldsymbol{r}}(\boldsymbol{\theta})^\top\boldsymbol{y}_i\right)^2\left(\tilde{\boldsymbol{w}}(\boldsymbol{\theta}; \boldsymbol{v})^\top\boldsymbol{y}_i\right)^2}{\frac{1}{n}\sum_{i=1}^{n}\left(\tilde{\boldsymbol{r}}(\boldsymbol{\theta})^\top\boldsymbol{y}_i\right)^2 \cdot \frac{1}{n}\sum_{i=1}^{n}\left(\tilde{\boldsymbol{w}}(\boldsymbol{\theta}; \boldsymbol{v})^\top\boldsymbol{y}_i\right)^2}.$$

One the one hand, with the help of Lemma C.4, there exists $C_1 > 0$ such that if we choose $n \gtrsim d + \log(1/\delta)$, then *w.p.* at least $1 - \delta/2$, we have:

$$\inf_{\boldsymbol{w},\boldsymbol{u}\in\mathbb{S}^{d-1}}\frac{1}{n}\sum_{i=1}^{n}(\boldsymbol{w}^\top\boldsymbol{y}_i)^2(\boldsymbol{u}^\top\boldsymbol{y}_i)^2 \ge C_1.$$

On the other hand, with the help of Lemma C.3, if we choose $\epsilon = 1/2$ and $n \gtrsim d + \log(1/\delta)$, then *w.p.* at least $1 - \delta/2$, we have:

$$\sup_{\boldsymbol{w}\in\mathbb{S}^{d-1}}\frac{1}{n}\sum_{i=1}^{n}(\boldsymbol{w}^\top\boldsymbol{y}_i)^2 \ge 1 + \frac{1}{2} = \frac{3}{2},$$

Combining these two bounds, we obtain that: if we choose $\epsilon = 1/2$ and $n \gtrsim d + \log(1/\delta)$, then *w.p.* at least $1 - \delta$, we have:

$$\inf_{\boldsymbol{w}, \boldsymbol{u} \in \mathbb{S}^{d-1}} \frac{\frac{1}{n} \sum_{i=1}^n (\boldsymbol{w}^\top \boldsymbol{y}_i)^2 (\boldsymbol{u}^\top \boldsymbol{y}_i)^2}{\frac{1}{n} \sum_{i=1}^n (\boldsymbol{w}^\top \boldsymbol{y}_i)^2 \cdot \frac{1}{n} \sum_{i=1}^n (\boldsymbol{u}^\top \boldsymbol{y}_i)^2}$$

$$\geq \frac{\inf_{\boldsymbol{w}, \boldsymbol{u} \in \mathbb{S}^{d-1}} \frac{1}{n} \sum_{i=1}^n (\boldsymbol{w}^\top \boldsymbol{y}_i)^2 (\boldsymbol{u}^\top \boldsymbol{y}_i)^2}{\left( \sup_{\boldsymbol{w} \in \mathbb{S}^{d-1}} \frac{1}{n} \sum_{i=1}^n (\boldsymbol{w}^\top \boldsymbol{y}_i)^2 \right)^2} \geq \frac{4 C_1}{9},$$

which implies that

$$\inf_{\boldsymbol{\theta}, \boldsymbol{v} \in \mathbb{R}^p} g(\boldsymbol{\theta}; \boldsymbol{v}) \geq \min \left\{ 1, \inf_{\boldsymbol{w}, \boldsymbol{u} \in \mathbb{S}^{d-1}} \frac{\frac{1}{n} \sum_{i=1}^n (\boldsymbol{w}^\top \boldsymbol{y}_i)^2 (\boldsymbol{u}^\top \boldsymbol{y}_i)^2}{\frac{1}{n} \sum_{i=1}^n (\boldsymbol{w}^\top \boldsymbol{y}_i)^2 \cdot \frac{1}{n} \sum_{i=1}^n (\boldsymbol{u}^\top \boldsymbol{y}_i)^2} \right\} \geq \min \left\{ 1, \frac{4 C_1}{9} \right\}.$$

$\square$

## C.2    PROOF OF THEOREM 4.3

We first need a few lemmas.

**Lemma C.5.** *Let* $\boldsymbol{y}_1, \cdots, \boldsymbol{y}_n \overset{\text{i.i.d.}}{\sim} \mathcal{N}(\boldsymbol{0}, \boldsymbol{I}_d)$. *If* $n \gtrsim d^2 + \log^2(1/\delta)$, *then w.p. at least* $1 - \delta$, *we have*

$$\sup_{\boldsymbol{v} \in \mathbb{S}^{d-1}} \frac{1}{n} \sum_{i=1}^n (\boldsymbol{y}_i^\top \boldsymbol{v})^4 \leq 8.$$

*Proof.* For $\mathbb{S}^{d-1}$, its covering number has the bound:

$$\left( \frac{1}{\rho} \right)^d \leq \mathcal{N}(\mathbb{S}^{d-1}, \rho) \leq \left( \frac{2}{\rho} + 1 \right)^d,$$

so there exist a $\rho$-net on $\mathbb{S}^{d-1}$: $\mathcal{V} \subset \mathbb{S}^{d-1}$, s.t. $|\mathcal{V}| \leq \left( \frac{2}{\rho} + 1 \right)^d$.

Step I. Bounding the term on the $\rho$-net.

For a fixed $\boldsymbol{v} \in \mathcal{V}$, due to $\boldsymbol{y}_i \overset{\text{i.i.d.}}{\sim} \mathcal{N}(\boldsymbol{0}, \boldsymbol{I}_d)$, we can verify $(\boldsymbol{y}_i^\top \boldsymbol{v})^4$ is sub-Weibull random variable:

$$\mathbb{E} \exp \left( \left( (\boldsymbol{y}_i^\top \boldsymbol{v})^4 \right)^{1/2} \right) = \mathbb{E} \exp \left( (\boldsymbol{y}_i^\top \boldsymbol{v})^2 \right) \lesssim 1,$$

which means that there exist an absolute constant $C_1 \geq 1$ s.t. $\left\| (\boldsymbol{y}_i^\top \boldsymbol{v})^4 \right\|_{\psi_{1/2}} \leq C_1$.

By the concentration inequality for Sub-Weibull distribution with $\beta = 1/2$ (Lemma E.5) and $\mathbb{E} \left[ (\boldsymbol{y}^\top \boldsymbol{v})^4 \right] = 3$, there exists an absolute constant $C_2 \geq 1$ s.t.

$$\mathbb{P} \left( \left| \frac{1}{n} \sum_{i=1}^n \left[ (\boldsymbol{y}_i^\top \boldsymbol{v})^4 \right] - 3 \right| > \phi(n; \delta) \right) \leq 2\delta,$$

where $\phi(n; \delta) = C_2 \left( \sqrt{\frac{\log(1/\delta)}{n}} + \frac{\log^2(1/\delta)}{n} \right)$. Applying a union bound over $\boldsymbol{v} \in \mathcal{V}$, we have:

$$\mathbb{P} \left( \exists \boldsymbol{v} \in \mathcal{V} s.t. \left| \frac{1}{n} \sum_{i=1}^n \left[ (\boldsymbol{y}_i^\top \boldsymbol{v})^4 \right] - 3 \right| > \phi(n; \delta) \right)$$

$$\leq \mathbb{P} \left( \bigcup_{\boldsymbol{v} \in \mathcal{V}} \left\{ \left| \frac{1}{n} \sum_{i=1}^n \left[ (\boldsymbol{y}_i^\top \boldsymbol{v})^4 \right] - 3 \right| > \phi(n; \delta) \right\} \right) \leq \sum_{\boldsymbol{v} \in \mathcal{V}} \mathbb{P} \left( \left| \frac{1}{n} \sum_{i=1}^n \left[ (\boldsymbol{y}_i^\top \boldsymbol{v})^4 \right] - 3 \right| > \phi(n; \delta) \right)$$

$$\leq 2|\mathcal{V}| \exp\left(-\frac{n}{C_2^2}\right) = 2\left(\frac{2}{\rho} + 1\right)^d \delta.$$

So *w.p.* at least $1 - 2\left(\frac{2}{\rho} + 1\right)^d \delta$, we have:

$$\max_{\boldsymbol{v} \in \mathcal{V}} \frac{1}{n} \sum_{i=1}^{n} \left[(\boldsymbol{y}_i^\top \boldsymbol{v})^4\right] \leq 3 + \phi(n; \delta).$$

Step II. Estimate the error of the $\rho$-net approximation.

For simplicity, we denote

$$P := \max_{\boldsymbol{v} \in \mathbb{S}^{d-1}} \frac{1}{n} \sum_{i=1}^{n} \left[(\boldsymbol{y}_i^\top \boldsymbol{v})^4\right] \quad , Q := \max_{\boldsymbol{v} \in \mathcal{V}} \frac{1}{n} \sum_{i=1}^{n} \left[(\boldsymbol{y}_i^\top \boldsymbol{v})^4\right].$$

Let $\boldsymbol{v} \in \mathbb{S}^{d-1}$ such that $\frac{1}{n} \sum_{i=1}^{n} \left[(\boldsymbol{y}_i^\top \boldsymbol{v})^4\right] = P$, then there exist $\boldsymbol{v}_0 \in \mathcal{V}$, s.t. $\|\boldsymbol{v} - \boldsymbol{v}_0\| \leq \rho$.

On the one hand,

$$\left|\frac{1}{n}\sum_{i=1}^{n}(\boldsymbol{y}_i^\top \boldsymbol{v})^4 - \frac{1}{n}\sum_{i=1}^{n}(\boldsymbol{y}_i^\top \boldsymbol{v}_0)^4\right| = \left|\frac{1}{n}\sum_{i=1}^{n}\left((\boldsymbol{y}_i^\top \boldsymbol{v})^4 - (\boldsymbol{y}_i^\top \boldsymbol{v}_0)^4\right)\right|$$

$$= \left|\frac{1}{n}\sum_{i=1}^{n}\left(\boldsymbol{y}_i^\top(\boldsymbol{v} - \boldsymbol{v}_0)\right)\left(\boldsymbol{y}_i^\top(\boldsymbol{v} + \boldsymbol{v}_0)\right)\left((\boldsymbol{y}_i^\top \boldsymbol{v})^2 + (\boldsymbol{y}_i^\top \boldsymbol{v}_0)^2\right)\right|$$

$$\leq \left|\frac{1}{n}\sum_{i=1}^{n}\left(\boldsymbol{y}_i^\top(\boldsymbol{v} - \boldsymbol{v}_0)\right)\left(\boldsymbol{y}_i^\top(\boldsymbol{v} + \boldsymbol{v}_0)\right)(\boldsymbol{y}_i^\top \boldsymbol{v})^2\right| + \left|\frac{1}{n}\sum_{i=1}^{n}\left(\boldsymbol{y}_i^\top(\boldsymbol{v} - \boldsymbol{v}_0)\right)\left(\boldsymbol{y}_i^\top(\boldsymbol{v} + \boldsymbol{v}_0)\right)(\boldsymbol{y}_i^\top \boldsymbol{v}_0)^2\right|$$

$$\leq \sqrt{\frac{1}{n}\sum_{i=1}^{n}\left(\boldsymbol{y}_i^\top(\boldsymbol{v} - \boldsymbol{v}_0)\right)^2\left(\boldsymbol{y}_i^\top(\boldsymbol{v} + \boldsymbol{v}_0)\right)^2}\left(\sqrt{\frac{1}{n}\sum_{i=1}^{n}(\boldsymbol{y}_i^\top \boldsymbol{v})^4} + \sqrt{\frac{1}{n}\sum_{i=1}^{n}(\boldsymbol{y}_i^\top \boldsymbol{v}_0)^4}\right)$$

$$\leq \sqrt[4]{\frac{1}{n}\sum_{i=1}^{n}\left(\boldsymbol{y}_i^\top(\boldsymbol{v} - \boldsymbol{v}_0)\right)^4}\sqrt[4]{\frac{1}{n}\sum_{i=1}^{n}\left(\boldsymbol{y}_i^\top(\boldsymbol{v} + \boldsymbol{v}_0)\right)^4}\left(\sqrt{\frac{1}{n}\sum_{i=1}^{n}(\boldsymbol{y}_i^\top \boldsymbol{v})^4} + \sqrt{\frac{1}{n}\sum_{i=1}^{n}(\boldsymbol{y}_i^\top \boldsymbol{v}_0)^4}\right)$$

$$\leq \|\boldsymbol{v} - \boldsymbol{v}_0\| P^{1/4} \|\boldsymbol{v} + \boldsymbol{v}_0\| P^{1/4}(\sqrt{P} + \sqrt{Q}) \leq 2\rho\sqrt{P}(\sqrt{P} + \sqrt{Q})$$

On the other hand,

$$\left|\frac{1}{n}\sum_{i=1}^{n}(\boldsymbol{y}_i^\top \boldsymbol{v})^4 - \frac{1}{n}\sum_{i=1}^{n}(\boldsymbol{y}_i^\top \boldsymbol{v}_0)^4\right| \geq P - \sum_{i=1}^{n}(\boldsymbol{y}_i^\top \boldsymbol{v}_0)^4 \geq P - Q.$$

Hence, we obtain

$$P - Q \leq 2\rho\sqrt{P}(\sqrt{P} + \sqrt{Q}),$$

which means that

$$P \leq \left(\frac{1}{1 - 2\rho}\right)^2 Q.$$

Step III. The bound for any $\boldsymbol{v} \in \mathbb{S}^{d-1}$.

Select $\rho = \frac{1}{2}(1 - \frac{1}{\sqrt{2}})$ and denote $\delta' = 2(\frac{2}{\rho} + 1)^d \delta$. And we choose $n \gtrsim d^2 + \log^2(1/\delta')$, which ensures $\phi(n; \delta) \leq 1$.

Then combining the results in Step I and Step II, we know that: *w.p.* at least $1 - \delta'$, we have:

$$\max_{\boldsymbol{v} \in \mathcal{V}} \frac{1}{n} \sum_{i=1}^n \left[ (\boldsymbol{y}_i^\top \boldsymbol{v})^4 \right] \le 3 + 1 = 4; \quad \max_{\boldsymbol{v} \in \mathbb{S}^{d-1}} \frac{1}{n} \sum_{i=1}^n \left[ (\boldsymbol{y}_i^\top \boldsymbol{v})^4 \right] \le 2 \max_{\boldsymbol{v} \in \mathcal{V}} \frac{1}{n} \sum_{i=1}^n \left[ (\boldsymbol{y}_i^\top \boldsymbol{v})^4 \right],$$

which means

$$\max_{\boldsymbol{v} \in \mathbb{S}^{d-1}} \frac{1}{n} \sum_{i=1}^n \left[ (\boldsymbol{y}_i^\top \boldsymbol{v})^4 \right] \le 2 \cdot 4 = 8.$$

$\square$

**Lemma C.6.** *Let $\boldsymbol{x}_1, \cdots, \boldsymbol{x}_n \overset{\text{i.i.d.}}{\sim} \mathcal{N}(\boldsymbol{0}, \boldsymbol{I}_d)$. For any $\epsilon, \delta \in (0, 1)$, if we choose*

$$n \gtrsim \max \left\{ \left( d^2 \log^2 (1/\epsilon) + \log^2(1/\delta) \right) / \epsilon, \left( d \log (1/\epsilon) + \log(1/\delta) \right) / \epsilon^2 \right\},$$

*then* w.p. *at least $1 - \delta$, we have:*

$$\sup_{\boldsymbol{w}, \boldsymbol{v} \in \mathbb{S}^{d-1}} \left| \frac{1}{n} \sum_{i=1}^n (\boldsymbol{w}^\top \boldsymbol{x}_i)^2 (\boldsymbol{v}^\top \boldsymbol{x}_i)^2 - \mathbb{E}\left[ (\boldsymbol{w}^\top \boldsymbol{x}_1)^2 (\boldsymbol{v}^\top \boldsymbol{x}_1)^2 \right] \right| \le \epsilon.$$

*Proof.* For $\mathbb{S}^{d-1}$, its covering number has the bound:

$$\left( \frac{1}{\rho} \right)^d \le \mathcal{N}(\mathbb{S}^{d-1}, \rho) \le \left( \frac{2}{\rho} + 1 \right)^d,$$

so there exist two $\rho$-nets on $\mathbb{S}^{d-1}$: $\mathcal{W} \subset \mathbb{S}^{d-1}$ and $\mathcal{V} \subset \mathbb{S}^{d-1}$, s.t.

$$|\mathcal{W}| \le \left( \frac{2}{\rho} + 1 \right)^d, \quad |\mathcal{V}| \le \left( \frac{2}{\rho} + 1 \right)^d.$$

Step I. Bounding the term on the $\rho$-net.

In this step, will estimate the term $\left| \frac{1}{n} \sum_{i=1}^n (\boldsymbol{w}^\top \boldsymbol{x}_i)^2 (\boldsymbol{v}^\top \boldsymbol{x}_i)^2 - \mathbb{E}\left[ (\boldsymbol{w}^\top \boldsymbol{x})^2 (\boldsymbol{v}^\top \boldsymbol{x})^2 \right] \right|$ for any $\boldsymbol{w} \in \mathcal{W}$ and $\boldsymbol{v} \in \mathcal{V}$.

For fixed $\boldsymbol{w} \in \mathcal{W}$ and $\boldsymbol{v} \in \mathcal{V}$, we denote $X_i^{\boldsymbol{w}, \boldsymbol{v}} := (\boldsymbol{w}^\top \boldsymbol{x}_i)^2 (\boldsymbol{v}^\top \boldsymbol{x}_i)^2$. We can verify $X_i$ is a sub-Weibull random variable with $\beta = 1/2$ (Definition E.4):

$$\mathbb{E}\left[ \exp\left( \left| (\boldsymbol{w}^\top \boldsymbol{x}_i)^2 (\boldsymbol{v}^\top \boldsymbol{x}_i)^2 \right|^{1/2} \right) \right] = \mathbb{E}\left[ \exp\left( |\boldsymbol{w}^\top \boldsymbol{x}_i| |\boldsymbol{v}^\top \boldsymbol{x}_i| \right) \right]$$

$$\le \mathbb{E}\left[ \exp\left( \frac{(\boldsymbol{w}^\top \boldsymbol{x}_i)^2 + (\boldsymbol{v}^\top \boldsymbol{x}_i)^2}{2} \right) \right] = \mathbb{E}\left[ \exp\left( \frac{(\boldsymbol{w}^\top \boldsymbol{x}_i)^2}{2} \right) \exp\left( \frac{(\boldsymbol{v}^\top \boldsymbol{x}_i)^2}{2} \right) \right]$$

$$\overset{\text{Lemma E.6}}{\le} \sqrt{\mathbb{E}\left[ \exp\left( (\boldsymbol{w}^\top \boldsymbol{x}_i)^2 \right) \right]} \cdot \sqrt{\mathbb{E}\left[ \exp\left( (\boldsymbol{v}^\top \boldsymbol{x}_i)^2 \right) \right]} \overset{\|(\boldsymbol{v}^\top \boldsymbol{x}_i)^2\|_{\psi_1} \le C_3}{\lesssim} 1,$$

which means that there exists an absolute constant $C_4 \ge 1$, s.t. $\|X_i^{\boldsymbol{w}, \boldsymbol{v}}\|_{\psi_{1/2}} \le C_4$. By the concentration inequality for Sub-Weibull distribution with $\beta = 1/2$ (Lemma E.5), there exists an absolute constant $C_5 \ge 1$, s.t.

$$\mathbb{P}\left( \left| \frac{1}{n} \sum_{i=1}^n X_i^{\boldsymbol{w}, \boldsymbol{v}} - \frac{1}{n} \sum_{i=1}^n \mathbb{E}\left[ X_i^{\boldsymbol{w}, \boldsymbol{v}} \right] \right| > \psi(n; \delta) \right) \le \delta.$$

where $\psi(n; \delta) = C_5 \left( \sqrt{\frac{\log(1/\delta)}{n}} + \frac{(\log(1/\delta))^2}{n} \right)$.

Applying an union bound over $\boldsymbol{w} \in \mathcal{W}$ and $\boldsymbol{v} \in \mathcal{V}$, we have:

$$\mathbb{P}\left(\exists \boldsymbol{w} \in \mathcal{W}, \boldsymbol{v} \in \mathcal{V}, \text{s.t.} \left|\frac{1}{n}\sum_{i=1}^{n} X_i^{\boldsymbol{w},\boldsymbol{v}} - \frac{1}{n}\sum_{i=1}^{n}\mathbb{E}\big[X_i^{\boldsymbol{w},\boldsymbol{v}}\big]\right| > \psi(n;\delta)\right)$$

$$\leq \mathbb{P}\left(\bigcup_{(\boldsymbol{w},\boldsymbol{v})\in\mathcal{W}\times\mathcal{V}}\left\{\exists \boldsymbol{w} \in \mathcal{W}, \boldsymbol{v} \in \mathcal{V}, \text{s.t.} \left|\frac{1}{n}\sum_{i=1}^{n} X_i^{\boldsymbol{w},\boldsymbol{v}} - \frac{1}{n}\sum_{i=1}^{n}\mathbb{E}\big[X_i^{\boldsymbol{w},\boldsymbol{v}}\big]\right| > \psi(n;\delta)\right\}\right)$$

$$\leq \sum_{(\boldsymbol{w},\boldsymbol{v})\in\mathcal{W}\times\mathcal{V}} \mathbb{P}\left(\exists \boldsymbol{w} \in \mathcal{W}, \boldsymbol{v} \in \mathcal{V}, \text{s.t.} \left|\frac{1}{n}\sum_{i=1}^{n} X_i^{\boldsymbol{w},\boldsymbol{v}} - \frac{1}{n}\sum_{i=1}^{n}\mathbb{E}\big[X_i^{\boldsymbol{w},\boldsymbol{v}}\big]\right| > \psi(n;\delta)\right)$$

$$\leq 2|\mathcal{W}||\mathcal{V}|\delta \leq 2\left(\frac{2}{\rho}+1\right)^{2d}\delta.$$

So *w.p.* at least $1 - 2\left(\frac{2}{\rho}+1\right)^{2d}\delta$, we have:

$$\sup_{\boldsymbol{w}\in\mathcal{W},\boldsymbol{v}\in\mathcal{V}}\left|\frac{1}{n}\sum_{i=1}^{n}(\boldsymbol{w}^\top\boldsymbol{x}_i)^2(\boldsymbol{v}^\top\boldsymbol{x}_i)^2 - \mathbb{E}\Big[(\boldsymbol{w}^\top\boldsymbol{x})^2(\boldsymbol{v}^\top\boldsymbol{x})^2\Big]\right| \leq \psi(n;\delta).$$

**Step II. Estimate the population error of the $\rho$-net approximation.**

Let $\boldsymbol{w}, \boldsymbol{v}, \boldsymbol{w}_0, \boldsymbol{v}_0 \in \mathbb{S}^{d-1}$, s.t. $\|\boldsymbol{w} - \boldsymbol{w}_0\| \leq \rho$ and $\|\boldsymbol{v} - \boldsymbol{v}_0\| \leq \rho$. For the population error, we have

$$\left|\mathbb{E}\Big[(\boldsymbol{w}^\top\boldsymbol{x})^2(\boldsymbol{v}^\top\boldsymbol{x})^2\Big] - \mathbb{E}\Big[(\boldsymbol{w}_0^\top\boldsymbol{x})^2(\boldsymbol{v}_0^\top\boldsymbol{x})^2\Big]\right|$$

$$= \left|\mathbb{E}\Big[\big((\boldsymbol{w}^\top\boldsymbol{x})^2 - (\boldsymbol{w}_0^\top\boldsymbol{x})^2\big)(\boldsymbol{v}^\top\boldsymbol{x})^2\Big] + \mathbb{E}\Big[(\boldsymbol{w}_0^\top\boldsymbol{x})^2\big((\boldsymbol{v}^\top\boldsymbol{x})^2 - (\boldsymbol{v}_0^\top\boldsymbol{x})^2\big)\Big]\right|$$

$$\leq \left|\mathbb{E}\Big[\big((\boldsymbol{w}^\top\boldsymbol{x})^2 - (\boldsymbol{w}_0^\top\boldsymbol{x})^2\big)(\boldsymbol{v}^\top\boldsymbol{x})^2\Big]\right| + \left|\mathbb{E}\Big[(\boldsymbol{w}_0^\top\boldsymbol{x})^2\big((\boldsymbol{v}^\top\boldsymbol{x})^2 - (\boldsymbol{v}_0^\top\boldsymbol{x})^2\big)\Big]\right|$$

We first bound $\left|\mathbb{E}\Big[\big((\boldsymbol{w}^\top\boldsymbol{x})^2 - (\boldsymbol{w}_0^\top\boldsymbol{x})^2\big)(\boldsymbol{v}^\top\boldsymbol{x})^2\Big]\right|$:

$$\left|\mathbb{E}\Big[\big((\boldsymbol{w}^\top\boldsymbol{x})^2 - (\boldsymbol{w}_0^\top\boldsymbol{x})^2\big)(\boldsymbol{v}^\top\boldsymbol{x})^2\Big]\right| = \left|\mathbb{E}\Big[\big((\boldsymbol{w} - \boldsymbol{w}_0)^\top\boldsymbol{x}\boldsymbol{x}^\top(\boldsymbol{w} + \boldsymbol{w}_0)(\boldsymbol{v}^\top\boldsymbol{x})^2\big)\Big]\right|$$

$$\leq \left(\mathbb{E}\left[\big((\boldsymbol{w} - \boldsymbol{w}_0)^\top\boldsymbol{x}\boldsymbol{x}^\top(\boldsymbol{w} + \boldsymbol{w}_0)\big)^2\right]\right)^{1/2}\left(\mathbb{E}\left[(\boldsymbol{v}^\top\boldsymbol{x})^4\right]\right)^{1/2}$$

$$\leq \left(\mathbb{E}\left[\big((\boldsymbol{w} - \boldsymbol{w}_0)^\top\boldsymbol{x}\big)^4\right]\right)^{1/4}\left(\mathbb{E}\left[\big((\boldsymbol{w} + \boldsymbol{w}_0)^\top\boldsymbol{x}\big)^4\right]\right)^{1/4}\left(\mathbb{E}\left[(\boldsymbol{v}^\top\boldsymbol{x})^4\right]\right)^{1/2}$$

$$\leq 3\|(\boldsymbol{w} - \boldsymbol{w}_0)\|\|(\boldsymbol{w} + \boldsymbol{w}_0)\|\|\boldsymbol{v}\|^2 \leq 6\rho.$$

Repeating the proof above, we also have:

$$\left|\mathbb{E}\Big[\big((\boldsymbol{w}^\top\boldsymbol{x})^2 - (\boldsymbol{w}_0^\top\boldsymbol{x})^2\big)(\boldsymbol{v}^\top\boldsymbol{x})^2\Big]\right| \leq 6\rho.$$

Combining these two inequalities, we have:

$$\left|\mathbb{E}\Big[(\boldsymbol{w}^\top\boldsymbol{x})^2(\boldsymbol{v}^\top\boldsymbol{x})^2\Big] - \mathbb{E}\Big[(\boldsymbol{w}_0^\top\boldsymbol{x})^2(\boldsymbol{v}_0^\top\boldsymbol{x})^2\Big]\right| \leq 6\rho + 6\rho = 12\rho.$$

Due to the arbitrariness of $\boldsymbol{w}, \boldsymbol{v}, \boldsymbol{w}_0, \boldsymbol{v}_0$, we obtain

$$\sup_{\substack{\boldsymbol{w},\boldsymbol{v},\boldsymbol{w}_0,\boldsymbol{v}_0\in\mathbb{S}^{d-1} \\ \|\boldsymbol{w}-\boldsymbol{w}_0\|\leq\rho,\|\boldsymbol{v}-\boldsymbol{v}_0\|\leq\rho}}\left|\mathbb{E}\Big[(\boldsymbol{w}^\top\boldsymbol{x})^2(\boldsymbol{v}^\top\boldsymbol{x})^2\Big] - \mathbb{E}\Big[(\boldsymbol{w}_0^\top\boldsymbol{x})^2(\boldsymbol{v}_0^\top\boldsymbol{x})^2\Big]\right| \leq 12\rho.$$

**Step III. Estimate the empirical error of the $\rho$-net approximation.**

Let $\boldsymbol{w}, \boldsymbol{v}, \boldsymbol{w}_0, \boldsymbol{v}_0 \in \mathbb{S}^{d-1}$, s.t. $\|\boldsymbol{w} - \boldsymbol{w}_0\| \leq \rho$ and $\|\boldsymbol{v} - \boldsymbol{v}_0\| \leq \rho$. For the empirical error, we have

$$\left| \frac{1}{n} \sum_{i=1}^n (\boldsymbol{w}^\top \boldsymbol{x}_i)^2 (\boldsymbol{v}^\top \boldsymbol{x}_i)^2 - \frac{1}{n} \sum_{i=1}^n (\boldsymbol{w}_0^\top \boldsymbol{x}_i)^2 (\boldsymbol{v}_0^\top \boldsymbol{x}_i)^2 \right|$$

$$= \left| \frac{1}{n} \sum_{i=1}^n \left[ \left( (\boldsymbol{w}^\top \boldsymbol{x}_i)^2 - (\boldsymbol{w}_0^\top \boldsymbol{x}_i)^2 \right) (\boldsymbol{v}^\top \boldsymbol{x}_i)^2 \right] + \frac{1}{n} \sum_{i=1}^n \left[ (\boldsymbol{w}_0^\top \boldsymbol{x}_i)^2 \left( (\boldsymbol{v}^\top \boldsymbol{x}_i)^2 - (\boldsymbol{v}_0^\top \boldsymbol{x}_i)^2 \right) \right] \right|$$

$$\leq \left| \frac{1}{n} \sum_{i=1}^n \left[ \left( (\boldsymbol{w}^\top \boldsymbol{x}_i)^2 - (\boldsymbol{w}_0^\top \boldsymbol{x}_i)^2 \right) (\boldsymbol{v}^\top \boldsymbol{x}_i)^2 \right] \right| + \left| \frac{1}{n} \sum_{i=1}^n \left[ (\boldsymbol{w}_0^\top \boldsymbol{x}_i)^2 \left( (\boldsymbol{v}^\top \boldsymbol{x}_i)^2 - (\boldsymbol{v}_0^\top \boldsymbol{x}_i)^2 \right) \right] \right|$$

We first bound $\left| \frac{1}{n} \sum_{i=1}^n \left[ \left( (\boldsymbol{w}^\top \boldsymbol{x}_i)^2 - (\boldsymbol{w}_0^\top \boldsymbol{x}_i)^2 \right) (\boldsymbol{v}^\top \boldsymbol{x}_i)^2 \right] \right|$:

$$\left| \frac{1}{n} \sum_{i=1}^n \left[ \left( (\boldsymbol{w}^\top \boldsymbol{x}_i)^2 - (\boldsymbol{w}_0^\top \boldsymbol{x}_i)^2 \right) (\boldsymbol{v}^\top \boldsymbol{x}_i)^2 \right] \right| = \left| \frac{1}{n} \sum_{i=1}^n \left[ \left( (\boldsymbol{w} - \boldsymbol{w}_0)^\top \boldsymbol{x}_i \boldsymbol{x}_i^\top (\boldsymbol{w} + \boldsymbol{w}_0) (\boldsymbol{v}^\top \boldsymbol{x}_i)^2 \right) \right] \right|$$

$$\leq 2\rho \sup_{\boldsymbol{u} \in \mathbb{S}^{d-1}} \frac{1}{n} \sum_{i=1}^n (\boldsymbol{x}_i^\top \boldsymbol{u})^4.$$

Repeating the proof above, we also have $\left| \frac{1}{n} \sum_{i=1}^n \left[ (\boldsymbol{w}_0^\top \boldsymbol{x}_i)^2 \left( (\boldsymbol{v}^\top \boldsymbol{x}_i)^2 - (\boldsymbol{v}_0^\top \boldsymbol{x}_i)^2 \right) \right] \right| \leq 2\rho \sup_{\boldsymbol{u} \in \mathbb{S}^{d-1}} \frac{1}{n} \sum_{i=1}^n (\boldsymbol{x}_i^\top \boldsymbol{u})^4$. Combining these two bounds, we have:

$$\left| \frac{1}{n} \sum_{i=1}^n (\boldsymbol{w}^\top \boldsymbol{x}_i)^2 (\boldsymbol{v}^\top \boldsymbol{x}_i)^2 - \frac{1}{n} \sum_{i=1}^n (\boldsymbol{w}_0^\top \boldsymbol{x}_i)^2 (\boldsymbol{v}_0^\top \boldsymbol{x}_i)^2 \right| \leq 4\rho \sup_{\boldsymbol{u} \in \mathbb{S}^{d-1}} \frac{1}{n} \sum_{i=1}^n (\boldsymbol{x}_i^\top \boldsymbol{u})^4.$$

Using Lemma C.5, if $n \gtrsim d^2 + \log^2(1/\delta')$, then *w.p.* at least $1 - \delta'/2$, we have $\sup_{\boldsymbol{u} \in \mathbb{S}^{d-1}} \frac{1}{n} \sum_{i=1}^n (\boldsymbol{x}_i^\top \boldsymbol{u})^4 \leq 8$.

Hence, *w.p.* at least $1 - \delta'/2$, we have

$$\left| \frac{1}{n} \sum_{i=1}^n (\boldsymbol{w}^\top \boldsymbol{x}_i)^2 (\boldsymbol{v}^\top \boldsymbol{x}_i)^2 - \frac{1}{n} \sum_{i=1}^n (\boldsymbol{w}_0^\top \boldsymbol{x}_i)^2 (\boldsymbol{v}_0^\top \boldsymbol{x}_i)^2 \right| \leq 32\rho.$$

Due to the arbitrariness of $\boldsymbol{w}, \boldsymbol{v}, \boldsymbol{w}_0, \boldsymbol{v}_0$, we obtain

$$\sup_{\substack{\boldsymbol{w}, \boldsymbol{v}, \boldsymbol{w}_0, \boldsymbol{v}_0 \in \mathbb{S}^{d-1} \\ \|\boldsymbol{w} - \boldsymbol{w}_0\| \leq \rho, \|\boldsymbol{v} - \boldsymbol{v}_0\| \leq \rho}} \left| \frac{1}{n} \sum_{i=1}^n (\boldsymbol{w}^\top \boldsymbol{x}_i)^2 (\boldsymbol{v}^\top \boldsymbol{x}_i)^2 - \frac{1}{n} \sum_{i=1}^n (\boldsymbol{w}_0^\top \boldsymbol{x}_i)^2 (\boldsymbol{v}_0^\top \boldsymbol{x}_i)^2 \right| \leq 32\rho.$$

Step IV. The bound for any $\boldsymbol{w}, \boldsymbol{v} \in \mathbb{S}^{d-1}$.

Combining the results in Step I, II, and II, we know that *w.p.* at least $1 - \frac{\delta'}{2} - (\frac{2}{\rho} + 1)^d$, we have

$$\sup_{\boldsymbol{w} \in \mathcal{W}, \boldsymbol{v} \in \mathcal{V}} \left| \frac{1}{n} \sum_{i=1}^n (\boldsymbol{w}^\top \boldsymbol{x}_i)^2 (\boldsymbol{v}^\top \boldsymbol{x}_i)^2 - \mathbb{E}\left[ (\boldsymbol{w}^\top \boldsymbol{x})^2 (\boldsymbol{v}^\top \boldsymbol{x})^2 \right] \right| \leq \psi(n; \delta),$$

$$\sup_{\substack{\boldsymbol{w}, \boldsymbol{v}, \boldsymbol{w}_0, \boldsymbol{v}_0 \in \mathbb{S}^{d-1} \\ \|\boldsymbol{w} - \boldsymbol{w}_0\| \leq \rho, \|\boldsymbol{v} - \boldsymbol{v}_0\| \leq \rho}} \left| \mathbb{E}\left[ (\boldsymbol{w}^\top \boldsymbol{x})^2 (\boldsymbol{v}^\top \boldsymbol{x})^2 \right] - \mathbb{E}\left[ (\boldsymbol{w}_0^\top \boldsymbol{x})^2 (\boldsymbol{v}_0^\top \boldsymbol{x})^2 \right] \right| \leq 12\rho,$$

$$\sup_{\substack{\boldsymbol{w}, \boldsymbol{v}, \boldsymbol{w}_0, \boldsymbol{v}_0 \in \mathbb{S}^{d-1} \\ \|\boldsymbol{w} - \boldsymbol{w}_0\| \leq \rho, \|\boldsymbol{v} - \boldsymbol{v}_0\| \leq \rho}} \left| \frac{1}{n} \sum_{i=1}^n (\boldsymbol{w}^\top \boldsymbol{x}_i)^2 (\boldsymbol{v}^\top \boldsymbol{x}_i)^2 - \frac{1}{n} \sum_{i=1}^n (\boldsymbol{w}_0^\top \boldsymbol{x}_i)^2 (\boldsymbol{v}_0^\top \boldsymbol{x}_i)^2 \right| \leq 32\rho.$$

Then for any $\boldsymbol{w}, \boldsymbol{v} \in \mathbb{S}^{d-1}$, there exists $\boldsymbol{w}_0 \in \mathcal{W}, \boldsymbol{v}_0 \in \mathcal{V}$ s.t. $\|\boldsymbol{w} - \boldsymbol{w}_0\| \leq \rho$ and $\|\boldsymbol{v} - \boldsymbol{v}_0\| \leq \rho$, so

$$\left| \frac{1}{n} \sum_{i=1}^n (\boldsymbol{w}^\top \boldsymbol{x}_i)^2 (\boldsymbol{v}^\top \boldsymbol{x}_i)^2 - \mathbb{E}\left[ (\boldsymbol{w}^\top \boldsymbol{x})^2 (\boldsymbol{v}^\top \boldsymbol{x})^2 \right] \right|$$

$$
= \left| \frac{1}{n} \sum_{i=1}^{n} (\boldsymbol{w}^\top \boldsymbol{x}_i)^2 (\boldsymbol{v}^\top \boldsymbol{x}_i)^2 - \frac{1}{n} \sum_{i=1}^{n} (\boldsymbol{w}_0^\top \boldsymbol{x}_i)^2 (\boldsymbol{v}_0^\top \boldsymbol{x}_i)^2 + \frac{1}{n} \sum_{i=1}^{n} (\boldsymbol{w}_0^\top \boldsymbol{x}_i)^2 (\boldsymbol{v}_0^\top \boldsymbol{x}_i)^2 \right.
$$

$$
\left. - \mathbb{E} \left[ (\boldsymbol{w}_0^\top \boldsymbol{x})^2 (\boldsymbol{v}_0^\top \boldsymbol{x})^2 \right] + \mathbb{E} \left[ (\boldsymbol{w}_0^\top \boldsymbol{x})^2 (\boldsymbol{v}_0^\top \boldsymbol{x})^2 \right] - \mathbb{E} \left[ (\boldsymbol{w}^\top \boldsymbol{x})^2 (\boldsymbol{v}^\top \boldsymbol{x})^2 \right] \right|
$$

$$
\leq \left| \frac{1}{n} \sum_{i=1}^{n} (\boldsymbol{w}^\top \boldsymbol{x}_i)^2 (\boldsymbol{v}^\top \boldsymbol{x}_i)^2 - \frac{1}{n} \sum_{i=1}^{n} (\boldsymbol{w}_0^\top \boldsymbol{x}_i)^2 (\boldsymbol{v}_0^\top \boldsymbol{x}_i)^2 \right|
$$

$$
+ \left| \frac{1}{n} \sum_{i=1}^{n} (\boldsymbol{w}_0^\top \boldsymbol{x}_i)^2 (\boldsymbol{v}_0^\top \boldsymbol{x}_i)^2 - \mathbb{E} \left[ (\boldsymbol{w}_0^\top \boldsymbol{x})^2 (\boldsymbol{v}_0^\top \boldsymbol{x})^2 \right] \right| + \left| \mathbb{E} \left[ (\boldsymbol{w}_0^\top \boldsymbol{x})^2 (\boldsymbol{v}_0^\top \boldsymbol{x})^2 \right] - \mathbb{E} \left[ (\boldsymbol{w}^\top \boldsymbol{x})^2 (\boldsymbol{v}^\top \boldsymbol{x})^2 \right] \right|
$$

$$
\leq \sup_{\substack{\boldsymbol{w}, \boldsymbol{v}, \boldsymbol{w}_0, \boldsymbol{v}_0 \in \mathbb{S}^{d-1} \\ \|\boldsymbol{w} - \boldsymbol{w}_0\| \leq \rho, \|\boldsymbol{v} - \boldsymbol{v}_0\| \leq \rho}} \left| \frac{1}{n} \sum_{i=1}^{n} (\boldsymbol{w}^\top \boldsymbol{x}_i)^2 (\boldsymbol{v}^\top \boldsymbol{x}_i)^2 - \frac{1}{n} \sum_{i=1}^{n} (\boldsymbol{w}_0^\top \boldsymbol{x}_i)^2 (\boldsymbol{v}_0^\top \boldsymbol{x}_i)^2 \right|
$$

$$
+ \sup_{\boldsymbol{w} \in \mathcal{W}, \boldsymbol{v} \in \mathcal{V}} \left| \frac{1}{n} \sum_{i=1}^{n} (\boldsymbol{w}^\top \boldsymbol{x}_i)^2 (\boldsymbol{v}^\top \boldsymbol{x}_i)^2 - \mathbb{E} \left[ (\boldsymbol{w}^\top \boldsymbol{x})^2 (\boldsymbol{v}^\top \boldsymbol{x})^2 \right] \right|
$$

$$
+ \sup_{\substack{\boldsymbol{w}, \boldsymbol{v}, \boldsymbol{w}_0, \boldsymbol{v}_0 \in \mathbb{S}^{d-1} \\ \|\boldsymbol{w} - \boldsymbol{w}_0\| \leq \rho, \|\boldsymbol{v} - \boldsymbol{v}_0\| \leq \rho}} \left| \mathbb{E} \left[ (\boldsymbol{w}^\top \boldsymbol{x})^2 (\boldsymbol{v}^\top \boldsymbol{x})^2 \right] - \mathbb{E} \left[ (\boldsymbol{w}_0^\top \boldsymbol{x})^2 (\boldsymbol{v}_0^\top \boldsymbol{x})^2 \right] \right|
$$

$$
\leq 32\rho + \psi(n; \delta) + 12\rho = 44\rho + \psi(n; \delta).
$$

Due to the arbitrariness of $\boldsymbol{w}, \boldsymbol{v}$, we have

$$
\sup_{\boldsymbol{w}, \boldsymbol{v} \in \mathbb{S}^{d-1}} \left| \frac{1}{n} \sum_{i=1}^{n} (\boldsymbol{w}^\top \boldsymbol{x}_i)^2 (\boldsymbol{v}^\top \boldsymbol{x}_i)^2 - \mathbb{E} \left[ (\boldsymbol{w}^\top \boldsymbol{x})^2 (\boldsymbol{v}^\top \boldsymbol{x})^2 \right] \right| \leq 44\rho + \psi(n; \delta)
$$

Select $\rho = \frac{\epsilon}{66}$ and $\delta'/2 = 2(1 + \frac{2}{\rho})^{2d}\delta$. And we choose

$$
n \gtrsim \max \left\{ \left( d^2 \log^2 (1/\epsilon) + \log^2 (1/\delta) \right) / \epsilon, (d \log (1/\epsilon) + \log(1/\delta)) / \epsilon^2 \right\},
$$

which satisfies $\psi(n; \delta) \leq \epsilon/3$.

Then *w.p.* at least $1 - \delta'/2 - \delta'/2 = 1 - \delta'$, we have

$$
\sup_{\boldsymbol{w}, \boldsymbol{v} \in \mathbb{S}^{d-1}} \left| \frac{1}{n} \sum_{i=1}^{n} (\boldsymbol{w}^\top \boldsymbol{x}_i)^2 (\boldsymbol{v}^\top \boldsymbol{x}_i)^2 - \mathbb{E} \left[ (\boldsymbol{w}^\top \boldsymbol{x})^2 (\boldsymbol{v}^\top \boldsymbol{x})^2 \right] \right| \leq \frac{44}{66}\epsilon + \frac{1}{3}\epsilon = \epsilon.
$$

$\square$

With the preparation of Lemma C.1, C.3, and C.6, now we give the proof of Theorem 4.3.

**Proof of Theorem 4.3.** Let $\boldsymbol{y}_i = \boldsymbol{S}^{-1/2} \boldsymbol{x}_i$, then $\boldsymbol{y}_1, \cdots, \boldsymbol{y}_n \overset{\text{i.i.d.}}{\sim} \mathcal{N}(\boldsymbol{0}, I_d)$.

$$
g(\boldsymbol{\theta}; \boldsymbol{v}) = \frac{\frac{1}{n} \sum\limits_{i=1}^{n} \left( \boldsymbol{r}^\top(\boldsymbol{\theta}) \boldsymbol{x}_i \right)^2 \left( (\nabla F(\boldsymbol{\theta}) \boldsymbol{v})^\top \boldsymbol{x}_i \right)^2}{\frac{1}{n} \sum\limits_{i=1}^{n} \left( \boldsymbol{r}^\top(\boldsymbol{\theta}) \boldsymbol{x}_i \right)^2 \cdot \frac{1}{n} \sum\limits_{i=1}^{n} \left( (\nabla F(\boldsymbol{\theta}) \boldsymbol{v})^\top \boldsymbol{x}_i \right)^2}
$$

$$
= \frac{\frac{1}{n} \sum\limits_{i=1}^{n} \left( (\boldsymbol{S}^{1/2} \boldsymbol{r}(\boldsymbol{\theta}))^\top \boldsymbol{y}_i \right)^2 \left( (\boldsymbol{S}^{1/2} \nabla F(\boldsymbol{\theta}) \boldsymbol{v})^\top \boldsymbol{y}_i \right)^2}{\frac{1}{n} \sum\limits_{i=1}^{n} \left( (\boldsymbol{S}^{1/2} \boldsymbol{r}(\boldsymbol{\theta}))^\top \boldsymbol{y}_i \right)^2 \cdot \frac{1}{n} \sum\limits_{i=1}^{n} \left( (\boldsymbol{S}^{1/2} \nabla F(\boldsymbol{\theta}) \boldsymbol{v})^\top \boldsymbol{y}_i \right)^2},
$$

Case (i). If $\boldsymbol{S}^{1/2} \boldsymbol{r}(\boldsymbol{\theta}) = \boldsymbol{0}$ or $\boldsymbol{S}^{1/2} \nabla F(\boldsymbol{\theta}) \boldsymbol{v} = 0$, we have $g(\boldsymbol{\theta}; \boldsymbol{v}) = \frac{0}{0} = 1$, this theorem holds.

Case (ii). If $\boldsymbol{S}^{1/2}\boldsymbol{r}(\boldsymbol{\theta}) \neq \boldsymbol{0}$ and $\boldsymbol{S}^{1/2}\nabla F(\boldsymbol{\theta})\boldsymbol{v} \neq 0$, we define the following normalized vectors:

$$\tilde{\boldsymbol{r}}(\boldsymbol{\theta}) := \frac{\boldsymbol{S}^{1/2}\boldsymbol{r}(\boldsymbol{\theta})}{\left\|\boldsymbol{S}^{1/2}\boldsymbol{r}(\boldsymbol{\theta})\right\|} \in \mathbb{S}^{d-1} \quad \tilde{\boldsymbol{w}}(\boldsymbol{\theta}; \boldsymbol{v}) := \frac{\boldsymbol{S}^{1/2}\nabla F(\boldsymbol{\theta})\boldsymbol{v}}{\left\|\boldsymbol{S}^{1/2}\nabla F(\boldsymbol{\theta})\boldsymbol{v}\right\|} \in \mathbb{S}^{d-1}.$$

From the homogeneity of $g(\boldsymbol{\theta}; \boldsymbol{v})$, we have:

$$g(\boldsymbol{\theta}; \boldsymbol{v}) = \frac{\frac{1}{n}\sum_{i=1}^{n}\left(\tilde{\boldsymbol{r}}(\boldsymbol{\theta})^{\top}\boldsymbol{y}_i\right)^2\left(\tilde{\boldsymbol{w}}(\boldsymbol{\theta}; \boldsymbol{v})^{\top}\boldsymbol{y}_i\right)^2}{\frac{1}{n}\sum_{i=1}^{n}\left(\tilde{\boldsymbol{r}}(\boldsymbol{\theta})^{\top}\boldsymbol{y}_i\right)^2 \cdot \frac{1}{n}\sum_{i=1}^{n}\left(\tilde{\boldsymbol{w}}(\boldsymbol{\theta}; \boldsymbol{v})^{\top}\boldsymbol{y}_i\right)^2}.$$

By Lemma C.3 and C.6, for any $\epsilon, \delta \in (0, 1)$, if we choose

$$n \gtrsim \max\left\{\left(d^2\log^2(1/\epsilon) + \log^2(1/\delta)\right)/\epsilon, \left(d\log(1/\epsilon) + \log(1/\delta)\right)/\epsilon^2\right\},$$

then *w.p.* at least $1 - \delta$, the following inequalities hold:

$$\sup_{\boldsymbol{v}\in\mathbb{S}^{d-1}}\left|\frac{1}{n}\sum_{i=1}^{n}(\boldsymbol{v}^{\top}\boldsymbol{y}_i)^2 - 1\right| \leq \epsilon,$$

$$\sup_{\boldsymbol{w},\boldsymbol{v}\in\mathbb{S}^{d-1}}\left|\frac{1}{n}\sum_{i=1}^{n}(\boldsymbol{w}^{\top}\boldsymbol{y}_i)^2(\boldsymbol{v}^{\top}\boldsymbol{y}_i)^2 - \mathbb{E}\left[(\boldsymbol{w}^{\top}\boldsymbol{y}_1)^2(\boldsymbol{v}^{\top}\boldsymbol{y}_1)^2\right]\right| \leq \epsilon;$$

These imply that for any $\boldsymbol{\theta}, \boldsymbol{v} \in \mathbb{R}^p$, we have:

$$\frac{\mathbb{E}\left[(\tilde{\boldsymbol{r}}(\boldsymbol{\theta})^{\top}\boldsymbol{y})^2(\tilde{\boldsymbol{w}}(\boldsymbol{\theta}; \boldsymbol{v})^{\top}\boldsymbol{y})^2\right] - \epsilon}{(1+\epsilon)^2} \leq g(\boldsymbol{\theta}; \boldsymbol{v}) \leq \frac{\mathbb{E}\left[(\tilde{\boldsymbol{r}}(\boldsymbol{\theta})^{\top}\boldsymbol{y}_1)^2(\tilde{\boldsymbol{w}}(\boldsymbol{\theta}; \boldsymbol{v})^{\top}\boldsymbol{y}_1)^2\right] + \epsilon}{(1-\epsilon)^2}. \tag{13}$$

First, we derive the upper bound for (13):

$$\mathrm{RHS} = \frac{\epsilon}{(1-\epsilon)^2} + \frac{\mathbb{E}\left[(\tilde{\boldsymbol{r}}(\boldsymbol{\theta})^{\top}\boldsymbol{y})^2(\tilde{\boldsymbol{w}}(\boldsymbol{\theta}; \boldsymbol{v})^{\top}\boldsymbol{y})^2\right]}{(1-\epsilon)^2\left(\tilde{\boldsymbol{r}}(\boldsymbol{\theta})^{\top}\tilde{\boldsymbol{r}}(\boldsymbol{\theta})\right)\left(\tilde{\boldsymbol{w}}(\boldsymbol{\theta}; \boldsymbol{v})^{\top}\tilde{\boldsymbol{w}}(\boldsymbol{\theta}; \boldsymbol{v})\right)}$$

$$\overset{\text{Homogeneity}}{=} \frac{\epsilon}{(1-\epsilon)^2} + \frac{\mathbb{E}\left[((\boldsymbol{S}^{1/2}\boldsymbol{r}(\boldsymbol{\theta}))^{\top}\boldsymbol{y})^2((\boldsymbol{S}^{1/2}\nabla F(\boldsymbol{\theta})\boldsymbol{v})^{\top}\boldsymbol{y})^2\right]}{(1-\epsilon)^2\left((\boldsymbol{S}^{1/2}\boldsymbol{r}(\boldsymbol{\theta}))^{\top}\boldsymbol{S}^{1/2}\boldsymbol{r}(\boldsymbol{\theta})\right)\left((\boldsymbol{S}^{1/2}\nabla F(\boldsymbol{\theta})\boldsymbol{v})^{\top}(\boldsymbol{S}^{1/2}\nabla F(\boldsymbol{\theta})\boldsymbol{v})\right)}$$

$$= \frac{\epsilon}{(1-\epsilon)^2} + \frac{\boldsymbol{v}^{\top}\Sigma(\boldsymbol{\theta})\boldsymbol{v}}{2(1-\epsilon)^2\mathcal{L}(\boldsymbol{\theta})\boldsymbol{v}^{\top}G(\boldsymbol{\theta})\boldsymbol{v}} \overset{\text{Lemma C.1}}{=} \frac{\epsilon}{(1-\epsilon)^2} + \frac{2\mathcal{L}(\boldsymbol{\theta})\boldsymbol{v}^{\top}G(\boldsymbol{\theta})\boldsymbol{v} + \left(\nabla\mathcal{L}(\boldsymbol{\theta})^{\top}\boldsymbol{v}\right)^2}{2(1-\epsilon)^2\mathcal{L}(\boldsymbol{\theta})\boldsymbol{v}^{\top}G(\boldsymbol{\theta})\boldsymbol{v}}$$

$$= \frac{1+\epsilon}{(1-\epsilon)^2} + \frac{\left(\nabla\mathcal{L}(\boldsymbol{\theta})^{\top}\boldsymbol{v}\right)^2}{2(1-\epsilon)^2\mathcal{L}(\boldsymbol{\theta})\boldsymbol{v}^{\top}G(\boldsymbol{\theta})\boldsymbol{v}} \overset{\text{Lemma C.2}}{\leq} \frac{1+\epsilon}{(1-\epsilon)^2} + \frac{1}{(1-\epsilon)^2} = \frac{2+\epsilon}{(1-\epsilon)^2}.$$

Moreover, if $\langle\boldsymbol{v}, \mathcal{L}(\boldsymbol{\theta})\rangle = 0$, then the bound is

$$\mathrm{RHS} \leq \frac{1+\epsilon}{(1-\epsilon)^2}.$$

In the similar way, we can derive the lower bound for (13):

$$\mathrm{LHS} = \frac{\boldsymbol{v}^{\top}\Sigma(\boldsymbol{\theta})\boldsymbol{v}}{2(1+\epsilon)^2\mathcal{L}(\boldsymbol{\theta})\boldsymbol{v}^{\top}G(\boldsymbol{\theta})\boldsymbol{v}} - \frac{\epsilon}{(1+\epsilon)^2} \overset{\text{Lemma C.1}}{=} \frac{2\mathcal{L}(\boldsymbol{\theta})\boldsymbol{v}^{\top}G(\boldsymbol{\theta})\boldsymbol{v} + \left(\nabla\mathcal{L}(\boldsymbol{\theta})^{\top}\boldsymbol{v}\right)^2}{2(1+\epsilon)^2\mathcal{L}(\boldsymbol{\theta})\boldsymbol{v}^{\top}G(\boldsymbol{\theta})\boldsymbol{v}} - \frac{\epsilon}{(1+\epsilon)^2}$$

$$\geq \frac{1}{(1+\epsilon)^2} - \frac{\epsilon}{(1+\epsilon)^2} = \frac{1-\epsilon}{(1+\epsilon)^2}.$$

So for any $\boldsymbol{S}^{1/2}\boldsymbol{u}(\boldsymbol{\theta}) \neq \boldsymbol{0}, \boldsymbol{S}^{1/2}\nabla F(\boldsymbol{\theta})\boldsymbol{v} \neq 0$, we have

$$\frac{1-\epsilon}{(1+\epsilon)^2} \leq g(\boldsymbol{\theta}; \boldsymbol{v}) \leq \frac{2+\epsilon}{(1-\epsilon)^2}.$$

Moreover, if $\langle v, \nabla \mathcal{L}(\theta) \rangle = 0$, then

$$\frac{1-\epsilon}{(1+\epsilon)^2} \leq g(\theta; v) \leq \frac{1+\epsilon}{(1-\epsilon)^2}.$$

Hence, we have proved this theorem: For any $\epsilon, \delta > 0$, if $n \gtrsim \max\left\{\left(d^2 \log^2(1/\epsilon) + \log^2(1/\delta)\right)/\epsilon, \left(d\log(1/\epsilon) + \log(1/\delta)\right)/\epsilon^2\right\}$, then $w.p.$ at least $1 - \delta$, the strong alignment holds uniformly:

(i). $\dfrac{1-\epsilon}{(1+\epsilon)^2} \leq \inf_{\theta, v \in \mathbb{R}^p} g(\theta; v) \leq \sup_{\theta, v \in \mathbb{R}^p} g(\theta; v) \leq \dfrac{2+\epsilon}{(1-\epsilon)^2}$,

(ii). $\dfrac{1-\epsilon}{(1+\epsilon)^2} \leq \inf_{\theta \in \mathbb{R}^p, \langle v, \nabla \mathcal{L}(\theta)\rangle = 0} g(\theta; v) \leq \sup_{\theta \in \mathbb{R}^p, \langle v, \nabla \mathcal{L}(\theta)\rangle = 0} g(\theta; v) \leq \dfrac{1+\epsilon}{(1-\epsilon)^2}$.

$\square$

# D PROOFS IN SECTION 5: ESCAPE DIRECTION OF SGD

## D.1 PROOF OF THEOREM 5.2

Recall that $w(t) = \sum_{i=1}^d w_i(t) u_i$ with $w_i(t) = u_i^\top w(t)$. Then, $w_i(t+1) = (1 - \eta\lambda_i)w_i(t) + \eta\xi(t)^\top u_i$. Taking the expectation of the square of both sides, we obtain

$$\mathbb{E}\left[w_i^2(t+1)\right] = (1 - \eta\lambda_i)^2 \mathbb{E}\left[w_i^2(t)\right] + \eta^2 \mathbb{E}[|u_i^\top \xi(t)|^2],$$

According to Assumption 5.1, there exists $A_1, A_2 > 0$ such that for any $i \in [d]$,

$$A_1 \lambda_i \mathcal{L}(w_t) \leq \mathbb{E}[|u_i^T \xi(t)|] \leq A_2 \lambda_i \mathcal{L}(w_t).$$

Let $X_t = \sum_{i=1}^k \lambda_i \mathbb{E}[w_i^2(t)], Y_t = \sum_{i=k+1}^d \lambda_i \mathbb{E}[w_i^2(t)]$ denote the components of loss energy along sharp and flat directions, respectively. And we denote $D_k(t) := Y_t/X_t$.

Plugging the fact that $2\mathcal{L}(w(t)) = X_t + Y_t$ into the two formulations above, we can obtain the following component dynamics:

$$X_{t+1} \leq \alpha_k X_t + A_2 \eta^2 \left(\sum_{i=1}^k \lambda_i^2\right)(X_t + Y_t),$$

$$X_{t+1} \geq A_1 \eta^2 \left(\sum_{i=1}^k \lambda_i^2\right)(X_t + Y_t), \tag{14}$$

$$Y_{t+1} \geq A_1 \eta^2 \left(\sum_{i=k+1}^d \lambda_i^2\right)(X_t + Y_t),$$

where $\alpha_k \leq \max_{i=1,\dots,k} |1 - \eta\lambda_i|^2$. The terms $\alpha_k X_t$ and $\beta_k Y_t$ capture the impact of the gradient, while the remaining terms originate from the noise.

From (14), we have the following estimate about $D_k(t+1)$:

$$D_k(t+1) = \frac{Y_{t+1}}{X_{t+1}} \geq \frac{A_1 \eta^2 \left(\sum_{i=k+1}^d \lambda_i^2\right)(X_t + Y_t)}{\alpha_k X_t + A_2 \eta^2 \left(\sum_{i=1}^k \lambda_i^2\right)(X_t + Y_t)}$$

$$= \frac{A_1 \sum_{i=k+1}^d \lambda_i^2}{A_2 \sum_{i=1}^k \lambda_i^2} \cdot \frac{1}{1 + \frac{\alpha_k}{A_2 \eta^2 \sum_{i=k+1}^d \lambda_i^2} \frac{X_t}{X_t + Y_t}} \tag{15}$$

$$\geq \frac{A_1 \sum_{i=k+1}^d \lambda_i^2}{A_2 \sum_{i=1}^k \lambda_i^2} \cdot \frac{1}{1 + \frac{\max\limits_{1 \leq i \leq k} |1-\eta\lambda_i|^2}{A_2 \eta^2 \sum_{i=1}^k \lambda_i^2} \frac{X_t}{X_t + Y_t}}.$$

We will prove this theorem for the learning rate $\eta = \frac{\beta}{\|G(\boldsymbol{\theta}^*)\|_F}$, where $\beta \geq \frac{1.1}{\sqrt{A_1}}$.

Case (I). Small learning rate $\eta \in \left[\frac{1.1}{\sqrt{A_1}\|G(\boldsymbol{\theta}^*)\|_F}, \frac{1}{\lambda_1}\right]$.

In this step, we consider $\eta = \frac{\beta}{\|G(\boldsymbol{\theta}^*)\|_F}$ such that $\beta \geq \frac{1.1}{\sqrt{A_1}}$ and $\eta \leq \frac{1}{\lambda_1}$. Then we have:

$$\frac{\max\limits_{1 \leq i \leq k} |1 - \eta\lambda_i|^2}{A_2\eta^2 \sum_{i=k+1}^{d} \lambda_i^2} \leq \frac{1}{A_2\eta^2 \sum_{i=1}^{k} \lambda_i^2}.$$

Notice that (14) also ensures:

$$(X_{t+1} + Y_{t+1}) \geq A_1\eta^2 \Big(\sum_{i=1}^{d} \lambda_i^2\Big)(X_t + Y_t).$$

Combining this inequality with (14), we have the estimate:

$$\frac{X_{t+1}}{X_{t+1} + Y_{t+1}} \leq \frac{\alpha_k X_t + A_2\eta^2(\sum_{i=1}^{k} \lambda_i^2)(X_t + Y_t)}{X_{t+1} + Y_{t+1}}$$

$$\leq \frac{\alpha_k X_t}{A_1\eta^2 \big(\sum_{i=1}^{d} \lambda_i^2\big)(X_t + Y_t)} + \frac{A_2(\sum_{i=1}^{k} \lambda_i^2)}{A_1\big(\sum_{i=1}^{d} \lambda_i^2\big)}$$

For simplicity, we denote $W_t := \frac{X_t}{X_t + Y_t}$, $A := \frac{\alpha_k}{A_1\eta^2\big(\sum_{i=1}^{d} \lambda_i^2\big)}$, and $B := \frac{A_2(\sum_{i=1}^{k} \lambda_i^2)}{A_1\big(\sum_{i=1}^{d} \lambda_i^2\big)}$.

From $\eta \leq 1/3$, we have $\alpha_k \leq 1$ and $A \leq \frac{1}{A_1\eta^2\big(\sum_{i=1}^{d} \lambda_i^2\big)} = \frac{1}{A_1\beta^2} < 1$. Moreover, it holds that

$$W_{t+1} \leq AW_t + B \leq A(AW_{t-1} + B) + B = A^2W_{t-1} + B(1 + A)$$

$$\leq \cdots \leq A^{t+1}W_0 + B(1 + A + \cdots + A^t) = A^{t+1}W_0 + \frac{1 - A^{t+1}}{1 - A}B$$

On the one hand, if we choose

$$t \geq \frac{\log\Big(1/W_0 A_2\eta^2 \sum_{i=1}^{k} \lambda_i^2\Big)}{\log\big(A_1\beta^2\big)},$$

then we have

$$A^t W_0 \leq \left(\frac{\alpha_k}{A_1\eta^2(\sum_{i=1}^{d} \lambda_i^2)}\right)^t W_0 \leq \left(\frac{1}{A_1\beta^2}\right)^t W_0 \leq A_2\eta^2 \sum_{i=1}^{k} \lambda_i^2.$$

On the other hand, if we choose $t \geq 1$, then it holds that

$$\frac{1 - A^t}{1 - A}B \leq B = \frac{A_2(\sum_{i=1}^{k} \lambda_i^2)}{A_1\big(\sum_{i=1}^{d} \lambda_i^2\big)} \leq A_2\eta^2 \sum_{i=1}^{k} \lambda_i^2.$$

Hence, if we choose

$$t \geq \max\left\{1, \frac{\log\Big(1/W_0 A_2\eta^2 \sum_{i=1}^{k} \lambda_i^2\Big)}{\log\big(A_1\beta^2\big)}\right\},$$

then we have

$$\frac{X_t}{X_t + Y_t} = W_t \leq A^t W_0 + \frac{1 - A^t}{1 - A}B \leq 2A_2\eta^2 \sum_{i=1}^{k} \lambda_i^2,$$

which implies that

$$\text{RHS of (15)} \geq \frac{A_1 \sum_{i=k+1}^d \lambda_i^2}{A_2 \sum_{i=1}^k \lambda_i^2} \cdot \frac{1}{1 + \frac{\max_{1 \leq i \leq k} |1-\eta\lambda_i|^2}{A_2 \eta^2 \sum_{i=1}^k \lambda_i^2} \frac{X_t}{X_t + Y_t}}$$

$$\geq \frac{A_1 \sum_{i=k+1}^d \lambda_i^2}{A_2 \sum_{i=1}^k \lambda_i^2} \cdot \frac{1}{1 + \frac{1}{A_2 \eta^2 \sum_{i=1}^k \lambda_i^2} \cdot 2 A_2 \eta^2 \sum_{i=1}^k \lambda_i^2} = \frac{A_1 \sum_{i=k+1}^d \lambda_i^2}{3 A_2 \sum_{i=1}^k \lambda_i^2}.$$

Case (II). Large learning rate $\eta \geq 1/\lambda_1$.

In this step, we consider $\eta \geq \frac{1}{\lambda_1}$. Then for any $t \geq 0$, we have:

$$\text{RHS of (15)} = \frac{A_1 \sum_{i=k+1}^d \lambda_i^2}{A_2 \sum_{i=1}^k \lambda_i^2} \cdot \frac{1}{1 + \frac{\alpha_k}{\sum_{i=k+1}^d \lambda_i^2} \frac{X_t}{X_t + Y_t}} \geq \frac{A_1 \sum_{i=k+1}^d \lambda_i^2}{A_2 \sum_{i=1}^k \lambda_i^2} \cdot \frac{1}{1 + \frac{\max_{i \in [k]} |1-\eta\lambda_i|^2}{A_2 \eta^2 \sum_{i=1}^k \lambda_i^2}}$$

$$\geq \frac{A_1 \sum_{i=k+1}^d \lambda_i^2}{A_2 \sum_{i=1}^k \lambda_i^2} \cdot \frac{1}{1 + \frac{\max\{1, |1-\eta\lambda_1|^2\}}{A_2 \eta^2 \sum_{i=1}^k \lambda_i^2}} \geq \frac{A_1 \sum_{i=k+1}^d \lambda_i^2}{A_2 \sum_{i=1}^k \lambda_i^2} \cdot \frac{1}{1 + \frac{1}{A_2}} = \frac{A_1 \sum_{i=k+1}^d \lambda_i^2}{(A_2 + 1) \sum_{i=1}^k \lambda_i^2}.$$

Combining Case (I) and (II), we obtain this theorem: If we choose the learning rate $\eta = \frac{\beta}{\|G(\boldsymbol{\theta})\|_F}$, where $\beta \geq \frac{1.1}{\sqrt{A_1}}$, then for any

$$t \geq \max \left\{ 1, \frac{\log \left( 1/W_0 A_2 \eta^2 \sum_{i=1}^k \lambda_i^2 \right)}{\log \left( A_1 \beta^2 \right)} \right\},$$

we have

$$D_k(t+1) \geq \frac{A_1 \sum_{i=k+1}^d \lambda_i^2}{\max\{3A_2, A_2+1\} \sum_{i=1}^k \lambda_i^2}.$$

$\square$

## D.2 PROOF OF PROPOSITION 5.3

Recall that $\boldsymbol{w}(t) = \sum_{i=1}^d w_i(t)\boldsymbol{u}_i$ with $w_i(t) = \boldsymbol{u}_i^\top \boldsymbol{w}(t)$. Then, for GD, $w_i(t+1) = (1-\eta\lambda_i)w_i(t)$, which implies:

$$w_i(t) = (1 - \eta\lambda_i)^t w_i(0).$$

Therefore, for $\eta = \beta/\lambda_1$ ($\beta > 2$), it holds that

$$D_1(t) = \frac{\sum_{i=2}^d \lambda_i w_i^2(t)}{\lambda_1 w_1^2(t)} = \frac{\sum_{i=2}^d \lambda_i (1-\eta\lambda_i)^{2t} w_i^2(0)}{\lambda_1 (1-\eta\lambda_1)^{2t} w_1^2(0)}.$$

$\square$

## E USEFUL INEQUALITIES

**Lemma E.1** (Bernstein's Inequality (Vershynin, 2018))**.** *Suppose $\{X_1, \cdots, X_n\}$ are independent sub-Exponential random variables with $\|X_i\|_{\psi_1} \leq K$. Then there exists an absolute constant $c > 0$ such that for any $t \geq 0$, we have:*

$$\mathbb{P}\left( \left| \frac{1}{n}\sum_{i=1}^n X_i - \frac{1}{n}\sum_{i=1}^n \mathbb{E}[X_i] \right| > t \right) \leq 2\exp\left( -cn\min\left\{ \frac{t}{K}, \frac{t^2}{K^2} \right\} \right).$$

**Lemma E.2** (Hanson-Wright's Inequality (Vershynin, 2018))**.** *Let $\boldsymbol{X} = (X_1, \cdots, X_n) \in \mathbb{R}^n$ be a random vector with independent mean zero sub-Gaussian coordinates. Let $\boldsymbol{A}$ be an $n \times n$ matrix. Then, there exists an absolute constant $c$ such that for every $t \geq 0$, we have*

$$\mathbb{P}\left(\left|\boldsymbol{X}^\top \boldsymbol{A} \boldsymbol{X} - \mathbb{E}[\boldsymbol{X}^\top \boldsymbol{A} \boldsymbol{X}]\right| \geq t\right) \leq 2 \exp\left(-c \min\left\{\frac{t^2}{K^4 \|\boldsymbol{A}\|_{\mathrm{F}}^2}, \frac{t}{K^2 \|\boldsymbol{A}\|_2}\right\}\right),$$

*where $K = \max_i \|X_i\|_{\psi_2}$.*

**Lemma E.3** (Covariance Estimate for sub-Gaussian Distribution (Vershynin, 2018))**.** *Let $\boldsymbol{x}, \boldsymbol{x}_1, \cdots, \boldsymbol{x}_n$ be i.i.d. random vectors in $\mathbb{R}^d$. More precisely, assume that there exists $K \geq 1$ s.t. $\|\langle \boldsymbol{x}, \boldsymbol{v} \rangle\|_{\psi_2} \leq K \|\langle \boldsymbol{x}, \boldsymbol{v} \rangle\|_{L_2}$ for any $\boldsymbol{v} \in \mathbb{S}^{d-1}$, Then for any $u \geq 0$, w.p. at least $1 - 2 \exp(-u)$ one has*

$$\left\|\frac{1}{n}\sum_{i=1}^n \boldsymbol{x}_i \boldsymbol{x}_i^\top - \mathbb{E}\left[\boldsymbol{x}\boldsymbol{x}^\top\right]\right\| \leq CK^2\left(\sqrt{\frac{d+u}{n}} + \frac{d+u}{n}\right)\left\|\mathbb{E}\left[\boldsymbol{x}\boldsymbol{x}^\top\right]\right\|,$$

*where $C$ is an absolute positive constant.*

**Definition E.4** (Sub-Weibull Distribution)**.** We define $X$ as a sub-Weibull random variable if it has a bounded $\psi_\beta$-norm. The $\psi_\beta$-norm of $X$ for any $\beta > 0$ is defined as

$$\|X\|_{\psi_\beta} := \inf\left\{C > 0 : \mathbb{E}\left[\exp(|X|^\beta / C^\beta)\right] \leq 2\right\}.$$

Particularly, when $\beta = 1$ or $2$, sub-Weibull random variables reduce to sub-Exponential or sub-Gaussian random variables, respectively.

**Lemma E.5** (Concentration Inequality for Sub-Weibull Distribution, Theorem 3.1 in (Hao et al., 2019))**.** *Suppose $\{X_i\}_{i=1}^n$ are independent sub-Weibull random variables with $\|X_i\|_{\psi_\beta} \leq K$. Then there exists an absolute constant $C_\beta$ only depending on $\beta$ such that for any $\delta \in (0, 1/e^2)$, w.p. at least $1 - \delta$, we have*

$$\left|\frac{1}{n}\sum_{i=1}^n X_i - \frac{1}{n}\sum_{i=1}^n \mathbb{E}[X_i]\right| \leq C_\beta K\left(\left(\frac{\log(1/\delta)}{n}\right)^{1/2} + \frac{\left(\log(1/\delta)\right)^{1/\beta}}{n}\right).$$

**Lemma E.6** (Cauchy-Schwarz Inequalities)**.**
*(1) Let $S \in \mathbb{R}^{n \times n}$ be a positive symmetric definite matrix. For any $\boldsymbol{x}, \boldsymbol{y} \in \mathbb{R}^n$, we denote $\langle \boldsymbol{x}, \boldsymbol{y} \rangle_S := \boldsymbol{x}^\top S \boldsymbol{y}$ and $\|\boldsymbol{x}\|_S := \sqrt{\langle \boldsymbol{x}, \boldsymbol{x} \rangle_S}$, then we have $|\langle \boldsymbol{x}, \boldsymbol{y} \rangle_S| \leq \|\boldsymbol{x}\|_S \|\boldsymbol{y}\|_S$.*
*(2) Given two random variables $X$ and $Y$, it holds that $|\mathbb{E}[XY]| \leq \sqrt{\mathbb{E}[X^2]}\sqrt{\mathbb{E}[Y^2]}$.*

