# OpenReview forum: "The Noise Geometry of Stochastic Gradient Descent: A Quantitative and Analytical Characterization"
_ICLR.cc/2024/Conference — ICLR 2024 Conference Withdrawn Submission_

### Official Review · Reviewer_anvg · 2023-10-30

**Soundness:** 3 good
**Presentation:** 3 good
**Contribution:** 3 good
**Rating:** 6
**Confidence:** 2

**Summary:**

This work investigates how the effective noise of stochastic gradient descent aligns with the landscape geometry, in particular with the Fisher matrix component of the Hessian of the empirical risk. It focus on Gaussian input data and two specific models: (a) overparametrised linear models and (b) two-layer neural networks with fixed second layer. The main results are high-probability lower and upper bounds for an alignement measure in different sample complexity regimes, as a function of the effective dimension of the data and architecture dependent quantities. The consequences for learning rate scheduling and for how SGD scapes sharp minima are discussed. Several numerical experiments with real networks trained on real data are provided, supporting the theoretical findings for the simplified setting.

**Strengths:**

- Understanding the SGD noise geometry is an important problem in theoretical ML, and the results provide some sharp answers for particular settings that are of interest to the NeurIPS theory community (OLMs and two-layer NNs).
- The paper is well written and easy to follow.
- The examples illustrating the assumptions are helpful, and provide intuition to the reader.
- The extensive comparison with real networks is welcome, and illustrate the relevance of the theoretical results.
- The authors are honest about the limitations of the theoretical results, highlighting the challenges and possible points of improvement.

**Weaknesses:**

While the mathematical results here are sound and the numerical experiment corroborate them beyond the theoretical scope, I miss a critical discussion of the claim that SGD noise helps escaping saddles/local minima. A couple of examples in the opposite direction can be found in the literature deriving exact scaling limits for SGD in the high-dimensional limit [Ben Arous et al. 2022; Arnaboldi et al 2023a]. For instance, in the phase retrieval problem it is known that SGD noise is effectively isotropic around the saddle-point at initialisation, and hence does not help escaping from it [Tan & Vershynin 2023; Arnaboldi et al. 2023b]. In some problems the effective SGD noise can even be degenerated around a saddle-point [Ben Arous et al. 2022].

**Questions:**

- **[Q1]** What are the target functions used in the synthetic experiments in Fig. 1(a,b)?
- **[Q2]** How essential is Assumption 3.3. for Theorem 3.5? For instance, for ReLU activation the lower bound becomes vacuous - is this a spurious limitation of the proof or the average alignement can be zero for two-layer ReLU networks? Have the authors run simulations in a synthetic setting?
- **[Q3]** I find somehow surprising that the lower bound on the alignement is not explicitly depending on the hidden layer width for two-layer neural networks in Theorem 3.5. For instance, [Arnaboldi et al 2023a] has shown that the effective noise contribution to the scaling limits of one-pass SGD is sub-leading in the hidden-layer width. Does the authors have any intuition for that discrepancy?

**Minor comments**

- Page 2: 'accors' -> 'across'
- Page 4: 'anistropic' -> 'anisotropic'
- Page 6: 'focues' -> 'focus'

**References**:
- **[Ben Arous et al. 2022]** Gerard Ben Arous, Reza Gheissari, Aukosh Jagannath. *High-dimensional limit theorems for SGD: Effective dynamics and critical scaling*. Part of Advances in Neural Information Processing Systems 35 (NeurIPS 2022).
- **[Arnaboldi et al 2023a]** Luca Arnaboldi, Ludovic Stephan, Florent Krzakala, Bruno Loureiro. *From high-dimensional & mean-field dynamics to dimensionless ODEs: A unifying approach to SGD in two-layers networks*. Proceedings of Machine Learning Research vol 195:1–29, 2023.
- **[Tan & Vershynin 2023]** Yan Shuo Tan, Roman Vershynin. *Online Stochastic Gradient Descent with Arbitrary Initialization Solves Non-smooth, Non-convex Phase Retrieval*. Journal of Machine Learning Research 24 (2023) 1-47.
- **[Arnaboldi et al. 2023b]** Luca Arnaboldi, Florent Krzakala, Bruno Loureiro, Ludovic Stephan. *Escaping mediocrity: how two-layer networks learn hard single-index models with SGD*. arXiv: 2305.18502 [stat.ML].

---

### Official Review · Reviewer_UF1v · 2023-10-31

**Soundness:** 2 fair
**Presentation:** 2 fair
**Contribution:** 1 poor
**Rating:** 3
**Confidence:** 4

**Summary:**

This paper provides an in-depth analysis of the gradient noise structure when training simple deep learning models. The authors, based on the analysis of Wu et al. (2022), provide results on the noise covariance for overparametrized linear models and 2 layer neural nets. The results cover general alignment as well as directional alignment. Finally, the authors use the investigated noise structure to show the inductive bias of SGD towards flat minimizers.

**Strengths:**

The paper is rigorous, and the topic needs investigation after the results of Wu et al. 2022. The motivation behind the paper is clear and I find the topic very interesting.

**Weaknesses:**

I unfortunately believe the structure, the results, and the experiments are a bit weak for acceptance. I think the authors should better organize the paper and provide more solid evidence that motivates publication after Wu et al. 2022

1) The structure is a bit hard to follow. The three main sections after page 3 feel somewhat separate. Though the leitmotiv is the gradient noise structure, I find no strong novel insights in the paper. Also, I found it a bit hard to get into the paper at first, since I was not aware of the results of Wu et al.: the authors transition at the end of page 3 directly to the setting of Wu et al. without giving the right intuition for studying the quantity of interest $\mu$. I had so many questions, but had to get back to the previous paper in order to "survive".

2) To my understanding, sections 3 and 4 provide a refinement of the results of Wu et al., while sections 5 and 6 show evidence of convergence to flat minimizers. The results, I would say, are not too surprising: they closely follow the claims of Wu et al. It's true that none of the precise bounds provided by the authors can be found in Wu et al., but the metric $\mu$ as well as the assumption the paper relies on, follow closely this work. In particular, the point of Wu et al. is also regularization towards flat minimizers, and they also show bounds on $\mu$. The authors should discuss how their results compare.

3) The experiments deal with the objects under investigation, so I am happy about the setting. However, take for example Figure 1: it is hard to get at first glance what is $p$ (one has to find the definition in between lines), and is not clear what the figure shows and how this result validates their claims: how does the empirical result compare to your bound? What does $\mu=1.2$ mean? I would find it much more insightful to compare $\Sigma$ with $LG$ in terms of relative L2 error.

4) Evasion from sharp minima: this section I think gives (as many other papers) the misconception that SGD "escapes" minima during training. I find this is very rarely the case, except if the starting point is planted. Do you have evidence of this in standard training?

**Questions:**

Some are above.

---

### Official Review · Reviewer_6rxs · 2023-11-01

**Soundness:** 2 fair
**Presentation:** 1 poor
**Contribution:** 3 good
**Rating:** 5
**Confidence:** 3

**Summary:**

This work studies the correlation between the noise of stochastic gradient descent and a proxy matrix based on the embedding, for over-parametrized linear models and two-layer neural networks, and shows that in certain regimes the correlation is at least bounded away from 0, if not nearly 1. Their results are in high probability and assume Gaussian data.  They then go on to investigate whether the noise is directionally aligned with the aforementioned proxy matrix, again for over-parametrized linear models with Gaussian data, and show that the directional alignment is bounded on one and two sides. Finally, they study the escape directions of SGD in a linearized setting and show that it prefers escape along flat directions. They offer empirics to support their theoretical results.

**Strengths:**

The authors develop many results and observations relating to the noise of stochastic gradient descent. They study both over-parametrized linear models and two layer neural networks, and consider both average and vector-wise alignment between the noise and the matrix $G(\theta)$. Their results seem to improve upon those in the literature as they apply for finite $n$ and more models. Their area of investigation is interesting both theoretically and practically.

**Weaknesses:**

My greatest concern with the paper is that it is often written in a very confusing way, with un-defined notation, un-explained figures, poorly emphasized/motivated assumptions, quantities that can be $0$ appearing in the denominators of fractions in Theorem statements, and little motivation for the results (see below for specific instances). Their theoretical results are limited to Gaussian data very simplistic models (over-parameterized linear models, 2 layer neural nets with the 2nd layer fixed), though I don't think this is a huge weakness since these phenomena are no doubt quite difficult to characterize in general. On the other hand the problem is really not well motivated - why do we care if the $G(\theta)$ and $\Sigma_1(\theta)$ matrices are well-aligned? The authors consider escape of SGD along flat vs sharp directions, but this doesn't seem to use their previous results on the noise alignment. They mention other applications like the edge of stability phenomenon but those are speculative. Finally, they offer lower bounds on the alignment of the matrices $G(\theta)$ and $\Sigma_1(\theta)$ but unless they also give upper bounds the lower bounds are not necessarily meaningful.

**Questions:**

- Is there a natural $O(1)$ upper bound on $\mu(\theta)$? Otherwise, proving that it is $\Omega(1)$ is not necessarily meaningful.
- I am concerned there is a math mistake in the proof of Theorem 3.1(a). In particular, at the last inequality on page 17, you seem to use the inequality  $\min_{i \in [n]} x_i^T ABA x_i \geqslant \lambda_{\rm min}(A)^2 \min_i x_i^T B x_i$ for $A = \nabla F(\theta) \nabla F(\theta)^T$ and $B = S$. Could you please justify this inequality? And if this isn't the inequality you are using, could you please explain this step in more detail?
- There are often typos in the first part of the proof of Theorem 3.1(a), such as when defining $y_i := S^{1/2}x_i$ (I believe this should be $y_i := S^{-1/2} x_i$).
- How is $\eta$ defined in Theorem 5.2 when $\|G(\theta^\star)\|_F= 0$?
- In the first part of Theorem 3.1, where $\log(n/\delta) \lesssim d_\{\rm eff} \varepsilon^2$, you say you are studying the "low-sample regime." I think it would help the presentation if you emphasize more that your results are independent of the dimension of the parameters, $p$. The dimension here is purely that of the data distribution. In particular, in the deep learning context we typically expect $d_{\rm eff} \ll n$, so the results for small $n$ are of less relevance.
- Could you give more details about  Figure 1? In particular, what $\theta$ are you measuring $\mu(\theta)$ for?
- The conditioning term in Theorem 3.1(a) should have an infimum over $\theta$
- It seems that the proofs of Theorem 3.1 are missing terms $y_i$ coming from the loss $\ell_i$. The proofs should still go through fine but please fix this omission.
- Figure 2 is confusing: you plot a quantity $\mu_k$, but don't say what it is, $\alpha_k$ is defined to be an expectation but is plotted as a random quantity, and each of these quantities is surely a function of $\theta$ yet it is not explained what $\theta$ value you selected to make the plots.
- I don't understand the statement of Theorem 5.2 at all. Firstly, what does it mean to "escape from that minima"? And secondly, is it saying that when $t$ is that large, the quantity $D_k(t)$ is large too?
- You frequently refer to $G$ is a "Hessian" matrix, for example at the top of page 6 in "Numerical validations" or section 5 after Theorem 5.2. But $G$ isn't a Hessian matrix, it is an outer product of gradients. Could you please clear this up?
- In section 5, why is the assumption that $\mathcal{L}(w) = \frac12 w^T G(\theta^\star) w$ well-motivated? It seems like even in the linear regime the loss may not equal this.

---

### Official Review · Reviewer_GyvU · 2023-11-04

**Soundness:** 3 good
**Presentation:** 3 good
**Contribution:** 1 poor
**Rating:** 3
**Confidence:** 3

**Summary:**

This paper examines the noise geometry of stochastic gradient descent. Precisely in the context of this paper, it refers to a metric as follows:

$\mu(\theta) = \frac{Tr(\Sigma_1(\theta) G(\theta))}{2\mathcal{L} (\theta) ||G(\theta)||_F^2} $.

Above,

- $\Sigma_1(\theta)$ is the averaged Jacobian (times Jacobian transpose) matrix.
- $G(\theta)$ is the sample covariance matrix.
- $\mathcal{L}(\theta)$ is the empirical risk.

(R1) This is called the "average alignment." The first result of this paper considers an overparametrized linear model, assuming the inputs are sampled from a normal distribution.
- The result shows a lower bound on the above average alignment metric, which scale inversely to the condition number of weight matrix of the linear model.
- The result gives a fine-grained dependence from $\mu$ to the sample size $n$.

(R2) The next result of this paper considers two-layer neural network setting, whose activation function is differentiable and whose derivative is bounded, given inputs sampled from a Gaussian distribution.
- This result similarly provides a lower bound on the alignment metric of $\mu(\theta)$, which depends on the bounds on the derivatives.

Next, the paper considers another metric called the directional alignment, stated as follows:

$g(\theta; v) = \frac{v^{\top} \Sigma(\theta) v} {2 L(\theta) (v^{\top} G(\theta) v)},$

where $\Sigma(\theta)$ is the centered Jacobian matrix.

(R1') The next result gives a lower bound on the above directional alignment metric in overparametrized linear models.

(R3) Lastly, this paper gives results on the "escape behavior" of SGD from sharp minima. This is contrasted with GD.

**Strengths:**

This paper provides a meticulous study of the geometry of solutions found by SGD, focusing on overparametrized linear models and two-layer neural networks.

The novelty can be justified as there is now a growing line of work on examining the flatness and sharpness of solutions found by optimization algorithms.

Experiments measure the directional alignment of several convolutional neural networks. The escape direction of SGD Is also illustrated.

**Weaknesses:**

Detailed comments on the results:

(R1) From a technical aspect, this result can be derived from regression analysis.

Additionally, the argument toward justifying $\mu(\theta)$ as the alignment metric is opaque.

Lastly, it is unclear to readers if the bound is tight or not.

Taken together, this result is not very convincing due to the above three issues.

(R2) The result on two-layer neural networks is by utilizing the Lipschitz smoothness of the activation.

It would be much more interesting if the result could be extended to multi-layer neural networks.

In the current form, the result looks preliminary.

(R3) This result is more interesting than the first two results in the referee's opinion, which characterizes a certain geometric measure for SGD and contrasts it with GD.

The result is again a bit preliminary in the referee's opinion. For instance, the result does not say how "fast" SGD will escape. The authors might want to check out a recent manuscript, which gives a convergence rate of a modified SGD algorithm to local minima penalized by the trace of the Hessian.

Ahn, Kwangjun, Ali Jadbabaie, and Suvrit Sra. "How to escape sharp minima." arXiv preprint arXiv:2305.15659 (2023).

Other comments:

There is a large body of work in overparametrized linear models. The related work discussion is incomplete. See the following examples and references therein:

Bartlett, P. L., Long, P. M., Lugosi, G., & Tsigler, A. (2020). Benign overfitting in linear regression. Proceedings of the National Academy of Sciences, 117(48), 30063-30070.

Tarmoun, Salma, et al. "Understanding the dynamics of gradient flow in overparameterized linear models." International Conference on Machine Learning. PMLR, 2021.

**Questions:**

See detailed comments above.